# Airborne measurements of fire Emission Factors for African biomass burning sampled during the MOYA Campaign

Patrick A. Barker.[1], Grant Allen.[1], Martin Gallagher.[1], Joseph R. Pitt.[2], Rebecca E. Fisher.[3], Thomas Bannan.[1], Euan G. Nisbet.[3], Stéphane J. -B. Bauguitte.[4], Dominika Pasternak.[5], Samuel Cliff.[5], Marina B. Schimpf.[6], Archit Mehra.[1], Keith N. Bower.[1], James D. Lee.[5], Hugh Coe.[1], Carl J. Percival.[7]

[1]School of Earth and Environmental Sciences, University of Manchester, Manchester, M13 9PL, UK.

[2]School of Marine and Atmospheric Sciences, 145 Endeavour Hall, Stony Brook University, Stony Brook, NY 11794-5000, USA.

[3]Department of Earth Sciences, Royal Holloway, University of London, Egham, Surrey, TW20 0EX, UK.

[4]FAAM Airborne Laboratory, National Centre for Atmospheric Sciences, Building 146, College Road, Cranfield, MK43 0AL, UK.

[5]Wolfson Atmospheric Chemistry Laboratories, Department of Chemistry, University of York, Heslington, York YO10 5DD, UK.

[6]German Aerospace Center (DLR), Flight Experiments, Instrumentation and Data Science, Muenchener Strasse 20, 82234 Wessling, Germany.

[7]Jet Propulsion Laboratory, California Institute of Technology, 4800 Oak Grove Drive, M/S 183-901, Pasadena, California 91109, USA.

*Correspondence to:* Patrick A. Barker (patrick.barker@manchester.ac.uk)

**Abstract.** Airborne sampling of methane ($CH_4$), carbon dioxide ($CO_2$), carbon monoxide (CO), and nitrous oxide ($N_2O$) mole fractions was conducted during field campaigns targeting fires over Senegal in February and March 2017, and Uganda in January 2019. The majority of fire plumes sampled were close to, or directly over burning vegetation, with the exception of two longer-range flights over the West African Atlantic seaboard, (100 – 300 km from source) where the continental outflow of biomass burning emissions from a wider area of West Africa was sampled. Fire Emission Factors (EFs) and modified combustion efficiencies (MCEs) were estimated from the enhancements in measured mole fractions. For the Senegalese fires, mean EFs and corresponding uncertainties in units of g per kg of dry fuel were $1.8 \pm 0.19$ for $CH_4$, $1633 \pm 171.4$ for $CO_2$ and $67 \pm 7.4$ for CO, with a mean MCE of $0.94 \pm 0.005$. For the Ugandan fires, mean EFs were $3.1 \pm 0.35$ for $CH_4$, $1610 \pm 169.7$ for $CO_2$ and $78 \pm 8.9$ for CO, with a mean modified combustion efficiency of $0.93 \pm 0.004$. A mean $N_2O$ EF of $0.08 \pm 0.002$ g $kg^{-1}$ is also reported for one flight over Uganda; issues with temperature control of the instrument optical bench prevented $N_2O$ EFs from being obtained for other flights over Uganda. This study has provided new datasets of African biomass burning EFs

and MCEs for two distinct study regions, in which both have been studied little by aircraft measurement previously. These results highlight the important intracontinental variability of biomass burning trace gas emissions, and can be used to better constrain future biomass burning emission budgets. More generally, these results highlight the importance of regional and fuel-type variability when attempting to spatially scale biomass burning emissions. Further work to constrain EFs at more local scales and for more specific (and quantifiable) fuel types will serve to improve global estimates of biomass burning emissions of climate-relevant gases.













## 1. Introduction

The atmospheric burdens of the greenhouse gases (GHGs) $CO_2$, $CH_4$ and $N_2O$ have been increasing since the onset of the Industrial Revolution. It is widely accepted that this increase is driven by anthropogenic emissions arising from rapid industrialisation and socio-economic development (Montzka et al. 2011; Ciais et al. 2013). However, there is significant uncertainty about the budgets of these greenhouse gases, as their sources and sinks, both natural and anthropogenic, remain poorly constrained. In particular, the continued growth in atmospheric methane since a period of stagnation from 1999-2006, alongside the concurrent shift in $^{13}CH_4/^{12}CH_4$ isotopic ratio, has yet to be accounted for (Nisbet et al. 2016, 2019; Turner et al. 2019; Schaefer. 2019). In order to accurately attribute the causes of the growth in greenhouse gas burdens, whether from increased sources or reduced sinks, all emission sources need to be quantified with accuracy and precision, and with fine detail in temporal and spatial variability.

Biomass burning is a major source, known to contribute significantly to the global budgets of many atmospheric trace gases and aerosols. In addition to $CO_2$, incomplete combustion of biomass fuel produces both methane and CO, as well as $N_2O$. It has been estimated that 1.6–4.1 Pg of $CO_2$, 11–53 Tg $CH_4$ and 0.1–0.3 Tg of $N_2O$ are emitted to the atmosphere annually as a result of biomass burning on a global scale (Crutzen and Andreae. 2016). The contribution of biomass burning to global GHG budgets will likely increase over time due to climate warming and more widespread drought-stress conditions which increase the likelihood and spread of wildfire events (Liu et al. 2014).

It is estimated that Africa accounts for approximately 52% of all biomass burning carbon emissions, with the Northern Sub-Saharan African region alone accounting for 20-25% of global biomass burning carbon emissions (van der Werf et al. 2010; Ichoku et al. 2016). Many or most of these fires are anthropogenic in origin and are started deliberately for reasons such as clearing land for agricultural use, crop waste burning, management of natural savannah vegetation, or as pest control (Andreae. 1991). Other fires may simply be accidental (e.g. cigarette disposal). Anthropogenic fires are typically lit in the winter dry season. Natural fires, lit by lightning, can occur in the first early summer wet season thunderstorms over dry growth from the previous year. Despite the importance of the African contribution to global biomass burning emissions, there are limited in situ studies of African wildfire emissions.

The UK Natural Environment Research Council (NERC) Methane Observations and Yearly Assessments (MOYA) project is focused primarily on closing the global methane budget through new in situ observations and analysis of existing datasets. This is being achieved (in part) through targeted field campaigns to constrain poorly-quantified methane sources on local and regional scales, as well as the use of atmospheric chemical transport models, such as GEOS-CHEM, to provide global estimates

of methane emission trends (Bey et al. 2001; Holmes et al. 2013; Saunois et al. 2016).

This paper presents the results of airborne surveys conducted over regions of Senegal and Uganda with high prevalence of biomass burning events. Two aircraft-based field campaigns, using the UK Facility for Airborne Atmospheric Measurements Atmospheric Research Aircraft (FAAM ARA), were conducted in widely separated parts of Northern Sub-Saharan Africa as part of the MOYA project. The first was based in Senegal between 27 February 2017 and 3 March 2017, and the second based in Uganda between 16 January 2019 and 30 January 2019 (henceforth referred to as MOYA-I and MOYA-II for the 2017 and 2019 campaigns respectively).

The primary focus of the Senegal campaign was to study fires in the winter dry season. The focus in the Ugandan campaign, which was carried out in the brief January dry season, was on equatorial wetlands, with the aim of quantifying methane emissions from these sources using regional-scale flux techniques (O'Shea et al. 2014; Heimburger et al. 2017), but the study of fires of opportunity in the savannah of Northern Uganda was also a major target. The aircraft campaigns also aimed to provide emission estimates for methane and other trace gas and aerosol species from other sources, including anthropogenic emissions from Kampala.

In particular, Emission Factors (EFs) for $CH_4$, $CO_2$, $N_2O$ and CO can be determined from the enhancement in trace gas mixing ratio observed when a biomass burning plume was intercepted. These EFs were calculated for multiple fires observed in Senegal and Uganda. A comparison is made between these Senegalese and Ugandan EFs, to assess and interpret intracontinental variability. Comparisons are also made between EFs determined in this study and EFs from Andreae (2019), who includes up to 50 studies reporting fire EFs and modified combustion efficiencies from multiple biomass burning types, such as tropical forest burning, savannah and grassland burning and agricultural residue burning.

## 2. Description of Flights and Experimental Methods

### 2.1. MOYA-I: Senegal 2017

During the first MOYA flying campaign (MOYA-I), four research flights (flight numbers C004, C005, C006 and C007) were conducted using the UK Facility for Airborne Atmospheric Measurement (FAAM) BAe 146-301 Atmospheric Research Aircraft (ARA) to specifically sample fire plumes from biomass burning. The ARA was based in Dakar for the duration of this flying campaign. Near-field biomass burning plumes were sampled in C004 and C005 above the Casamance region of wooded savannah in the south-west of Senegal, and longer-range biomass burning outflow for a wider West African region were sampled in C006 and C007 over the Atlantic seaboard.

Fig. 1 shows the NASA MODerate Resolution Imaging Spectrometer (MODIS) satellite retrievals of locations that were actively burning during the MOYA-I fire sampling flights, which both took place between 28 February 2017 and 2 March 2017. Several straight-and-level (constant altitude and heading) runs were made in the central Casamance region of south-west Senegal, to sample near-field biomass burning emissions from directly above the source fires. Straight and level runs were also carried out during flights C006 and C007 but aimed to sample longer-range regional outflow of biomass burning emissions from the wider inland area of interest.

Visual observation during low passes (<200m) in the flight showed that the fires were in wooded savannah terrain, in winter-dry and winter-brown forest tracts. The forests have been described by de Wolf. (1998) and by Fredericksen et al. (1992).
The likely fuels were C3 forest leaf litter and dropped branches as well as savannah grass. The Casamance forests in the overflown area were typically low trees with a generally open canopy. A photograph of one of the near-field fires sampled during flight C005 is shown in Fig. 2

### 2.2. MOYA-II: Uganda 2019

The flying campaign in Uganda (MOYA-II) took place in late January 2019, a relatively dry month, when northern Uganda experiences its winter dry season, and equatorial southern Uganda is in a short January dry period. The aircraft was based at Entebbe, located on the equator. Two dedicated biomass burning sampling research flights were conducted (flight numbers C133 conducted on 28 January 2019 and C134 conducted on 29 January 2019), which targeted burning occurring in the north-west of Uganda. Fig. 1 shows the flight tracks and MODIS-retrieved fire locations for the MOYA-II flights. The fires were

concentrated towards the north of Uganda in this period.

Fig. 1 shows both dedicated biomass burning sampling flights (C133 and C134), which focussed on the north-western corner of Uganda. This region is far enough north (around $3^{\circ}$N) to experience dry season northern-hemisphere winter. A box pattern was flown around the region, including several passes downwind of fires in the area seen with the clover-like flight patterns. In addition to these dedicated fires flights, flight C132 (conducted on 28 January 2019) is also included in emission analyses. This flight was over Lake Kyoga, closer to the equator at about 1.5°N. The primary purpose of flight C132 was to survey biogenic methane emissions from Lake Kyoga and the surrounding wetlands. Flight C132 involved straight-and-level runs across Lake Kyoga. No fires were specifically targeted during this flight but plumes were intercepted from fires over the northern area of Lake Kyoga, as seen by the deviations in the C132 flight path shown in Fig. 1. EFs from these fires are included in this study.

From visual observation, flights C133 and C134 likely included fires mainly burning C4 tropical grasses, and on flight C132 the fuel was likely agricultural crop waste, which presumably included C4 maize waste, a major local crop.

## 2.3. CH$_4$, CO$_2$, CO and N$_2$O Instrumentation

During the MOYA-I and MOYA-II campaigns, the FAAM ARA was equipped with a suite of instrumentation for high-accuracy and precision trace gas measurement. All airborne trace gas measurements are time synchronized to an on-board time server For CH$_4$ and CO$_2$ mole fractions, a Los Gatos Research Fast Greenhouse Gas Analyser (FGGA) was used. This instrument uses a Cavity-Enhanced Absorption Spectroscopy technique and two continuous-wave near-IR diode lasers. A more detailed description of this instrument, along with its modification for airborne measurements, is provided by O'Shea et al (2013). The FGGA was calibrated using 3 calibration gas standards, all of which were traceable to the NOAA/ESRL WMO-X2007 scale for CO$_2$, and the WMO-X2004A scale for CH$_4$. Two of these gas standards provide high/low-concentration span calibrations that are linearly interpolated over an entire flight in order to account for instrument drift. The remaining gas standard was used as a target to define instrumental measurement uncertainty across multiple flights. During MOYA-I the FGGA had a data acquisition rate of 1 Hz, whereas in MOYA-II we used an upgraded system with a 10 Hz acquisition rate. Accounting for all sources of uncertainty associated with these instruments, the mean biases and associated 1σ overall uncertainties are estimated to be $0.004 \pm 0.431$ ppm and $0.04 \pm 2.27$ ppb for 1 Hz CO$_2$ and CH$_4$ measurements respectively during MOYA-I, and $-0.048 \pm 0.626$ ppm and $-1.22 \pm 2.93$ ppb respectively for 10 Hz CO$_2$ and CH$_4$ measurements during MOYA-II, which have been averaged to 1 Hz prior to analysis.

N$_2$O dry-air mole fractions were measured using an Aerodyne Quantum Cascade Laser Absorption Spectrum (QCLAS) as described by Pitt et al. (2016). This instrument uses a single thermoelectrically cooled quantum cascade laser tuned to a wavelength of ~4.5 µm. The QCLAS is calibrated using three calibration gas standards, all of which are traceable to the World Meteorological Organisation (WMO) X2006 calibration scale for N$_2$O. A 1σ uncertainty of 0.58 ppb was estimated for 1 Hz N$_2$O mole fraction measurements during the MOYA-II flights. We only report data for the MOYA-II (Uganda) campaign in

this study as this instrument was not fitted to the aircraft during the MOYA-I (Senegal) campaign.

The Aerodyne QCLAS N$_2$O measurements can be impacted by changes in both cabin pressure and aircraft motion. Changes in altitude and hence cabin pressure change the refractive index in the open path section of the laser beam. This leads to normally static optical fringes moving across the spectral baseline of the instrument, introducing both long-term drift and short-term

artefacts into the N$_2$O mole fraction data. Sharp changes in aircraft roll angle in tight turns also introduce short-term artefacts as forces acting on optical components cause slight changes in alignment. These issues are described in further detail in Pitt et al. (2016). A further issue encountered solely during the MOYA-II campaign was occasional loss of optical bench temperature control due to the high temperatures experienced within the aircraft during some flights.

Despite these issues, the N$_2$O plumes from which EF could be calculated were sampled at constant altitude with wings-level at constant optical bench temperature. So the instrument issues detailed likely have a minimal influence on data quality during these periods.

Measurements of CO dry-air mole fractions were sampled using an AeroLaser AL5002 Vacuum-UV fast fluorescence

instrument. Specifics about the principles of operation for this instrument are provided by Gerbig et al., 1999. The instrument was calibrated in-flight using a gas standard traceable to the NOAA/ESRL WMO-X2014A scale for CO. We have demonstrated that the linear interpolation of in-flight calibrations yields a mean bias <1 ppb with a 2 sigma precision of 1.8 ppb at 150 ppb for 1 Hz CO measurements, when the instrument is operated optimally. However we recently discovered that a faulty inlet drier may have impacted the accuracy of our CO measurements in 2017-19, and yielded a +9 ± 9 ppb bias in our

data. The potential impact of this positive bias is further discussed.

Both the Aerolaser CO instrument and the FGGA were mounted within the pressurised cabin of the aircraft within a single 19" rack. Air was sampled by means of a window-mounted rearward facing inlet comprising of 3/8" PFA tubing housed within 1/2" stainless steel tubing for the CO inlet, and 3/8" stainless steel tubing for the FGGA inlet (O'Shea et al., 2013; Gerbig et

al., 1999).

## 2.4. HCN and HNCO Instrumentation (Chemical Ionisation Mass Spectrometer)

The University of Manchester Time of Flight Chemical Ionisation Mass Spectrometer (ToF-CIMS) that has been described in detail by Priestley et al. (2018a; 2018b) for ground-based deployment has recently been modified and certified for use on the FAAM ARA and was used for real time detection of hydrogen cyanide (HCN) and isocyanic acid (HNCO) in this study. The instrument and its subsequent modification is described in detail here, as this study presents the first measurements from the modified ToF-CIMS aboard the FAAM ARA. The original instrument was manufactured by Aerodyne Research Inc. and

employs the ARI/Tofwerk High Resolution Time of Flight Mass Spectrometer. Briefly, iodide ions cluster with sample gasses creating a stable adduct that is analysed using time of flight mass spectrometry, with an average mass resolution of 4000 (m/Δm).

      The inlet design was based on the configuration characterised by Le Breton et al. (2015), an atmospheric pressure, rearward

facing, short residence time inlet, consisting of a 3/8” diameter polytetrafluoroethylene (PTFE) tubing with a total length to the instrument of 48 cm and based on the design shown in Lee et al. (2018). A constant flow of 12 SLM is mass flow controlled to the ion-molecule reaction region (IMR) using a rotary vane pump (Picolino VTE-3). 1 SLM is then subsampled into the IMR for measurement. An Iris system as described by Lee et al. (2018) was then employed to pressure and mass flow control the sample flow into the instrument, avoiding sensitivity changes that would be associated with large variations in pressures in

flight that is not controlled sufficiently by the constant flow inlet. This works upon the principle of the manipulation of the size of the critical orifice in response to changes in the IMR pressure. As with the Lee et al. (2018) design, this works by having a stainless-steel plate with a critical orifice and a movable PTFE plate on top of this, also with a critical orifice. These orifices either align fully and allow maximum flow into the instrument or misalign to reduce flow. This movement is controlled by the 24VDC output of the IMR Pirani pressure gauge in relation to the set point and the control unit was designed collaboratively

with Aerodyne Research Inc. The IMR set point was 80 mbar for the MOYA campaign, which is set through a combination of pumping capacity on the region (Agilent IDP3), mass flow-controlled reagent ion flow and sample flow. The reagent ion flow is 1 SLM of ultra-high purity (UHP) nitrogen mixed with 2 SCCM of a pressured known concentration gas mix of $CH_3I$ in nitrogen, passed through the radioactive source, $^{210}$Po. The total flow through the IMR is measured (MKS MFM) at the exhaust of the Agilent IDP3 pump so that not only the IMR pressure is monitored but the sample flow also. All mass flow controllers

and mass flow meters are measured and controlled using EyeOn. The 1σ variability in the IMR pressure during MOYA is 4% and 6% in the sample flow.

      A standard Aerodyne pressure controller is also employed on the short segmented quadrupole (SSQ) region, with two purposes, easily setting the required pressure during start up but to also make subtle adjustments in this region should the IMR pressure

change significantly. This works upon the principle controlling an electrically actuated solenoid valve in a feedback loop with

the SSQ pressure gauge to actively control a leak of air into the SSQ pumping line. The SSQ is pumped using Ebara PDV 250 pump and held at 1.8 mbar. The $1\sigma$ variability in the SSQ pressure during MOYA is <1%.

Instrument backgrounds are programmatically run for 6 seconds every minute for the entire flight, by overflowing the inlet at
the point of entry into the IMR with UHP nitrogen. Here a 1/16th PTFE line enters through the movable PTFE top plate, ensuring that the flow exceeds that of the sample flow. Inlet backgrounds are often run multiple times during flights manually by overflowing as close to the end of the inlet as possible with 20 SLM. Data is taken at 4Hz during a flight, which is routinely averaged to 1 Hz for analysis. Of the 6 points in each background, the first 2 and last point are unused and the mean of the background is calculated using custom python scripting. Using linear interpolation, a time series of the instrument background
is determined, humidity corrected if required and then subtracted to give the final time series of each measured mass. Instrument sensitivity to increased humidity changes influences the sensitivity of the instrument to HCN and corrections are applied here to correct both the instrumental backgrounds and final time series of HCN reported here. Only qualitative HCN and HNCO data is reported here as quantitative data is not required for the approach of plume identification used in this study.

The CIMS instrument analysis software (ARI Tofware version 3.1.0) was utilized to attain high resolution, 1Hz, time series of the compounds presented here. For the UMan CIMS, mass-to-charge calibration was performed for 5 known masses; I-, I-.H$_2$O, I-.HCOOH, I$_2$-, I$_3$-, covering a mass range of 127 to 381 m/z. The mass-to-charge calibration was fitted to a 3rd order polynomial and was accurate to within 2 ppm. HCN and HNCO in this case were identified with a 1 ppm error.

**2.5 Whole Air Sampling and Methane Isotopic Analysis**

Whole air sample (WAS) were collected onboard the aircraft in 3L silica passivated stainless steel canisters (Thames Restek, UK). Sample collection was triggered manually to sample within and outside of fire plumes, guided by the real time methane measurements from the FGGA onboard and visual identification of when the plumes were being crossed. Fill times when
sampling the fire plumes ranged between 10 and 40 seconds depending on sampling altitude, representative of an integrated air sample over a 1 - 4 km track. WAS sampling start and end times are recorded using the time on the FAAM ARA on-board time server. Methane mole fraction in the WAS flasks was measured in the Royal Holloway greenhouse gas laboratory using a Picarro 1301 cavity ringdown spectroscopy analyser, and methane isotopic analysis ($\delta^{13}$C) was carried out by gas chromatography – isotope ratio mass spectrometry using a Trace Gas preconcentrator and Isoprime mass spectrometer (see
Fisher et al., 2006 for details of the technique).

**2.6 Calculation of Emission Ratios and Emission Factors**


In order to select when sampled air was influenced by biomass burning emissions, HCN and CO were used as biomass burning tracers. HCN was chosen as it is almost exclusively emitted from biomass burning, representing 70-85% of the total global HCN source (Li et al. 2003) and has a sufficiently long atmospheric lifetime (relative to advection timescales prior to sampling) of 2-4 months, making HCN a suitable inert tracer for characterising biomass burning plumes (Li et al. 2000).


Like HCN, significant amounts of CO, which has an atmospheric lifetime of 1-3 months (Ehhalt et al. 2001), are emitted from biomass burning. CO is also emitted by vehicles, primarily petrol-fuelled and less so by diesel. However, it is likely that biomass burning is the dominant source of carbonaceous emissions in rural areas of Africa as studied here, whereas vehicular carbon emissions are likely concentrated towards urban centres (Gatari et al. 2003). HCN was used as a biomass burning tracer

for the MOYA-II (Uganda) analysis. However, as the ToF-CIMS was not fitted to the aircraft during the MOYA-I campaign, no HCN measurement is available for this dataset, and hence CO is used as the biomass burning tracer for MOYA-I analysis.

In order to quantify biomass burning emissions from the enhancements in trace gas mole fraction seen in fire plumes, Emission Ratios (ERs) and EFs were calculated for each species in each fire plume. In this case, an ER is defined as the ratio of a species X relative to a reference species Y. The reference species chosen for this work was CO, as it is relatively inert in the timescale

of these measurements, had a relatively stable regional background concentration during these campaigns, and in these rural field areas is almost exclusively emitted during combustion processes and not by other sources such as vehicles (Andreae and Merlet. 2001). The expression for ER calculation is shown in Eq. (1).


$$ER_{\frac{X}{CO}} = \frac{\Delta X}{\Delta CO} = \frac{X_{plume} - X_{background}}{CO_{plume} - CO_{background}} \tag{1}$$

ERs calculated using this approach are also referred to as Normalised Excess Mixing Ratios (NEMRs). When fresh plumes are sampled close to source as they are in the near-field sampling flights, NEMRs can be treated as ERs, calculated using Eq.

(1). However in aged plumes, this approach cannot be used to calculate ER, and NEMR is no longer equal to ER. This is due both to chemical processes within the plume that can change composition as well as mixing of background air into plume air (Andreae and Merlet. 2001; O'Shea et al. 2013; Yokelson et al. 2013). HYSPLIT back-trajectory analysis of the MOYA-I far-field flights show that the plume age is <12 hours for flight C006, hence chemical ageing of biomass burning emissions is unlikely to significantly impact the ER calculation for this flight. Plume ages during flight C007 are more variable, and can

exceed 2 days in some cases, so significant ageing may have occurred. This is discussed in further detail in Sect. 3.2. All near-field flights sample biomass burning emissions at the source, so no significant plume ageing is assumed. Eq. (1) can therefore

be used to calculate ERs confidently for most flights.

In order to calculate ERs for near-field biomass burning plumes, a baseline mixing ratio ($X_{background}$) was calculated as the average mixing ratio over 10 seconds of sampled data to either side of each detected plume. The same baseline data periods chosen for each plume were used for all gas species, to ensure that ERs were comparable and not influenced by inconsistent baseline criteria. Plumes were selected using a statistical method, but the start and the end of each plume as well as the background regions were chosen manually. The area under the plume was then determined by integrating the peak in the concentration versus time data series, giving a total plume concentration ($X_{plume}$). These values were then used in Eq. (1), along

with the corresponding values for CO, to determine an ER. Due to the absence of individual sharp enhancements resolved for specific fire plumes in the far-field flights, a least-squares linear regression of all in-plume points of X versus in-plume points of CO is used to determine ERs for the far-field flights. The ER is equal to the slope of this linear regression.

Using the calculated ER for each species, EFs were calculated using the carbon mass balance technique (Ward et al. 1984;

Radke et al. 1988) An EF is defined as the mass of species emitted (in grams) per kilogram of dry matter burnt. The expression for calculating emission factor is given in Eq. (2).

$$EF_X = F_C \cdot 1000(g\ kg^{-1}) \cdot \frac{M_X}{M_c} \frac{C_X}{C_{total}} \tag{2}$$


where $F_C$ is the mass fraction of carbon in the dry fuel. A value of 0.475 was assumed in this work to best represent African biomass carbon content, and a $\pm 10\%$ uncertainty in this value is assumed (Cofer et al. 1996; Ward et al. 1996; Yokelson et al. 2009). $M_X$ is the molecular weight of species X and $M_C$ is the atomic mass of carbon-12. The term $\frac{C_X}{C_{total}}$ is the molar ratio of species X to total carbon in the plume, which is calculated using Eq. (3).


$$\frac{C_X}{C_{total}} = \frac{ER\frac{X}{CO}}{1+\frac{\Delta CO_2}{\Delta CO}+\frac{\Delta CH_4}{\Delta CO}} \tag{3}$$


In Eq. 3, total carbon in the fire plume was assumed to be the sum of CO, $CO_2$ and $CH_4$ emitted. However, as all carbon-containing species could not be measured in this study, the total carbon present in the plume may be underestimated by 1-2% (as reported by Yokelson et al. 1999).

A statistical threshold approach was used to determine when a biomass burning plume was sampled during flights. For flights where HCN measurements are available, HCN enhancements exceeding seven standard deviations above the local background were used to select data for ER and EF calculation. Where HCN was not available during MOYA-I, a CO threshold of seven standard deviations over the local background concentration was used. For the far-field flights during MOYA-I (C006 and C007) CO mixing ratios exceeding 15 standard deviations above the local background were chosen for analysis.


**2.7 Modified Combustion Efficiency**

In addition to EF, the modified combustion efficiency (MCE) is another useful parameter that can be calculated for each biomass burning plume. MCE is here defined by Eq. (4).


$$MCE = \frac{\Delta CO_2}{\Delta CO_2 + \Delta CO} \tag{4}$$


MCE can be used to determine the degree to which a fire is smouldering or flaming (Ward and Radke, 1993). Higher MCE values (towards 0.99) indicate that burning is purely flaming, whereas lower MCE values in the range 0.65-0.85 indicate that smouldering conditions dominate. The proportion of trace gases (such as CO and $CH_4$) emitted typically depends on the completeness of combustion, which is to say that more oxidised products are expected from fires with a high degree of flaming.

It is therefore useful to investigate the trend between EF and MCE for different fire plumes (Urbanski, 2013). In the following section, we calculate EFs and MCEs for sampled fire plumes in the MOYA-I and MOYA-II campaigns.

**2.8 Uncertainties**


The standard error of the mean (SE) and the mean measurement uncertainty (MU) are reported for each mean EF and MCE displayed in Table 1, The SE here is determined from all EF and MCE calculated for a single flight, and represents the variability of EF and MCE within a flight. The MU is propagated from the instrument uncertainties, therefore each EF and MCE from each fire plume sampled has a measurement uncertainty associated with it. The MUs reported in Table 1 are the

average of all individual MUs for all fire plumes sampled during a given flight.

$ER_x$ is calculated using Eq. (1) by subtracting $CO_{background}$ from $CO_{plume}$; any CO measurements systematic positive offset would therefore cancel out and not affect the uncertainty of $ER_x$. The detection of $CO_{plume}$ during MOYA-I is based on the exceedance of either seven or fifteen standard deviations above background; a CO measurements offset on the background may therefore affect this data filtering step; however due the wide dynamic range of CO measurements encountered during the plumes sampling, we believe a bias will have a very minimal effect on the filtered plume data set used in our analysis. Similarly, the calculations of $EF_x$ using Eq. (2) and (3), and MCE using Eq. (4), rely on $\Delta CO$, which is unaffected by CO measurement bias as previously stated.








## 3 Results & Discussion

In this section, mean EFs and MCEs are reported on a per-flight basis, and the differences in relative EFs and MCE between individual flights and between Senegal and Uganda are discussed.


### 3.1 Near-Field sampling

#### 3.1.1 MOYA-I

*Flights C004 and C005*

The near-field Senegalese fire sampling flights (flight C004 and C005) were carried out on 28 February 2017 and 1 March 2017 respectively. The operating area was over the south western Casamance region of Senegal. A time series of trace gas mixing ratios (CO, $CH_4$ and $CO_2$) during flight C004 is shown in Fig. 3. An equivalent time series for flight C005 is displayed in the supplementary information in Fig. S1

The $\delta^{13}C$-$CH_4$ isotopic ratio of biomass burning emissions can provide information on the content of the biomass fuel that is burned. In C4 vegetation (e.g. tropical grassland), $^{13}CO_2$ is concentrated during the photosynthetic pathway, hence C4 plants tend to be enriched in $^{13}C$ and emissions show a higher $\delta^{13}C$-$CH_4$ isotopic ratio. C3 vegetation (woody forest) does not involve the same $^{13}C$ fractionation as C4, therefore emissions show a lower $\delta^{13}C$-$CH_4$ ratio relative to C3 plants (Brownlow et al. 2017). Chanton et al. (2000) analysed biomass burning emissions via Keeling plot analysis ($\delta^{13}C$-$CH_4$ vs inverse $CH_4$ mole

fraction) from a range of fuel sources. They found that African grass burning emitted methane with $\delta^{13}C$-$CH_4$ ranging between -17 ‰ and -26 ‰, whereas African woodland burning produced methane with a $\delta^{13}C$-$CH_4$ ratio of approximately -30 ‰. For both near-field and far-field MOYA-I flights, whole-air samples were taken of the biomass burning plumes sampled, as well as of the local background. $\delta^{13}C$-$CH_4$ isotopic ratios and mean $CH_4$ mole fractions are determined from these whole-air samples. Further detail of this analysis is provided in Sect. 2.5. Keeling plots for all MOYA-I flights analysed in this work are

shown in Fig. 4. Flight C005 shows a linear relationship between inverse $CH_4$ mole fraction (enhanced $CH_4$) and $\delta^{13}C$-$CH_4$ signature. This suggests that biomass burning emissions were captured by whole-air sampling during flight C005 One sample taken during flight C004 appears to have an enriched $\delta^{13}C$-$CH_4$ signal, however this is not included in the linear fit as the single point does not conclusively mean a linear relationship is present. The Y-intercept of -33.7 $\pm$ 1.1 ‰ agrees well with the Chanton et al. (2000) estimate for African forest burning, and suggests that C3 vegetation (forest) is included in the fuel burned during

flight C005 (Dlugokencky et al. 2011, Chanton et al 2000). Unfortunately, flights over mixed sources in Uganda meant that Keeling plot analysis could not be used to determine the isotopic composition of fire emissions in the same way as carried out for Senegal. The Keeling plot for the MOYA-II isotope samples is shown in the supplementary information (Fig. S2)

Table 1 shows the EFs calculated for all species during flight C004 and C005, as well as savannah and grassland and tropical forest fire EFs reported by Andreae. (2019). The methane EFs for C004 and C005 ($2.3 \pm 0.24$ g kg$^{-1}$ and $1.4 \pm 0.15$ g kg$^{-1}$ respectively) in this region, at the northern fringe of the African moist tropics, is more comparable to the savannah and grassland methane EF ($2.7 \pm 2.2$ g kg$^{-1}$) averaged from multiple previous studies by Andreae. (2019). Additionally, mean CO EFs ($84 \pm 8.7$ g kg$^{-1}$ for C004 and $61 \pm 6.2$ g kg$^{-1}$ for C005) are also more comparable to the savannah and grassland CO EF of $69 \pm 20$ g kg$^{-1}$ than the tropical forest CO EF of $104 \pm 39$ g kg$^{-1}$ reported by Andreae. (2019).

The magnitude of methane EFs can be affected by multiple factors, such as fuel moisture (affecting combustion efficiency) as well as fuel type (Brownlow et al. 2017). It is worth noting that the majority of studies included in the Andreae (2019) tropical forest analysis focus on burning associated with Amazonian deforestation, which consists mostly of broad-leafed evergreen forest. In contrast, the Casamance region consists of facultatively deciduous broad-leafed forested savannah, which was observed from the aircraft and is shown by the land cover map in Fig. 5a. It is thus possible that any forest matter burned during the MOYA-I flights consists of dry leaf-litter fuel, whereas the Andreae (2019) study comprising mostly Amazonian land clearing may have included burning of whole evergreen tree structures. In addition to this, the modified combustion efficiencies of the C004 and C005 fires ($0.93 \pm 0.0031$ and $0.95 \pm 0.0030$ respectively) are both higher than that reported in Andreae (2019) for tropical forest ($0.91 \pm 0.03$), and are more comparable with the Andreae (2019) MCE for savannah and grassland burning ($0.94 \pm 0.02$). This is likely due to the lower fuel moisture content of dry leaf-litter and savannah grasses as opposed to Amazonian evergreen, hence the methane EFs are likely driven by combustion efficiency.

From the EF and $\delta^{13}$C-CH$_4$ results from flights C004 and C005, it is likely that the biomass fuel is a mixture of both deciduous forest matter and savannah grasses as inferred from the isotope and EF results, as well as visual observations of forested savannah and the presence of shrubland and open forest in the land cover classification (Fig. 5a).

### 3.1.2 MOYA-II

*Flight C132*

Flight C132 was undertaken on 28 January 2019, as a survey of the Lake Kyoga wetland area. Two crop waste biomass burning plumes were sampled from two distinct fires in the area (see Fig. 1). A time series of various trace gas mixing ratios during this flight is shown in Fig. 6.

As seen in Fig. 6, enhancements (relative to background) in all trace gases were observed in the two biomass burning plumes. However, N$_2$O mixing ratio data during the two enhancements were discarded due to aircraft turbulence, which may have corrupted data quality. As a result of the discarded data, as well as instrument drift owing to malfunction of the laser coolant system, N$_2$O EFs are not reported for flight C132.

Fig. 5b shows the land cover of Uganda where the fire sampling flights were carried out. In agreement with on-board observations from the aircraft, much of the land surrounding Lake Kyoga is classified as cropland, and the fuel for the fires

appeared to be primarily crop waste. This is a major farming region, with the main crops including maize (a C4 plant) and cassava (C3) south of Lake Kyoga, and sorghum (C4) north of the Lake. (FEWS NET. 2019). The mean EFs calculated for C132 ($5.2 \pm 0.55$ g kg$^{-1}$ for $CH_4$, $1554 \pm 164.2$ g kg$^{-1}$ for $CO_2$ and $109 \pm 11.3$ g kg$^{-1}$ for CO) agree within overlapping uncertainty with mean agricultural burning EFs of $5.7 \pm 6.0$ g kg$^{-1}$ for $CH_4$, $1430 \pm 240$ g kg$^{-1}$ for $CO_2$, and $76 \pm 55$ g kg$^{-1}$ for CO reported by Andreae (2019). The mean MCE obtained for the C132 fires ($0.90 \pm 0.0042$) is also in agreement with the Andreae (2019)

MCE for agricultural residue burning ($0.92 \pm 0.06$). Furthermore, compared to Northern Uganda, the Lake Kyoga region has a shorter dry season, and higher rainfall. In addition, the fires were bordering a wetland area. Thus the moister conditions of the Lake Kyoga fires could have resulted in lower temperature, moister combustion and therefore more incomplete burning.

*Flights C133 and C134*

Flights C133 and C134 were dedicated fire sampling flights surveying the winter savannah of north-west Uganda. Both flights involved box-patterns flown over this region, with deviations taken in order to sample biomass burning plumes visibly identified over the course of the flights. C133 and C134 were undertaken on 28 January 2019 and 29 January 2019 respectively. The trace gas time series for these flights are shown in the supplementary information in Fig. S3 and Fig. S4.

The $CH_4$, $CO_2$ and CO EFs determined for the fire plumes encountered during flight C133 ($2.8 \pm 0.30$) g kg$^{-1}$, $1620 \pm 171.2$) g kg$^{-1}$ and $72 \pm 7.4$ g kg$^{-1}$ respectively) agreed well with Andreae. (2019) savannah burning EFs ($2.7 \pm 2.2$ g kg$^{-1}$ for $CH_4$, $1660 \pm 90$ g kg$^{-1}$ for $CO_2$ and $69 \pm 20$ g kg$^{-1}$ for CO). The mean $CH_4$ and $CO_2$ EFs for C134 ($3.1 \pm 0.22$ and $1609 \pm 173.8$ g kg$^{-1}$ respectively) are broadly comparable with the $CH_4$ and $CO_2$ EFs calculated for C133. Additionally, the mean MCE for C134 ($0.93 \pm 0.0042$) is comparable to that of C133 ($0.94 \pm 0.0041$). The mean MCE for C133 and C134 demonstrate that the

burning observed in these flights was characterised by more complete flaming combustion than that observed in flight C132 ($0.90 \pm 0.0042$), resulting in the comparatively higher $CO_2$ EFs and lower $CH_4$ EFs determined for C133 and C134 relative to C132. The trends in mean MCE and EFs observed during C132, C133, and C134 suggest that EFs are mostly determined by the completeness of combustion over other factors, which is illustrated by the linear relationships between $CH_4$, $CO_2$ and CO EFs vs MCE shown in Fig. 7. In particular, fires sampled during C134 may have had a larger smouldering component and they

appeared to have involved less complete combustion on average than in C133, which would explain the lower emissions of more highly oxidised $CO_2$ and higher emissions of more reduced $CH_4$ than were observed in C134.

The ratio of HCN enhancement to HNCO enhancement within the plumes is informative to quantify combustion completeness and in order to provide redundancy in estimating fire combustion efficiency. Molar ratios of HCN to HNCO in fire emissions

have been shown to decrease linearly with increasing combustion temperature (Hansson et al., 2004). Hence lower $\Delta HCN/\Delta HNCO$ ratios should be expected from fires with more complete combustion. Fig. 8a shows $\Delta HCN/\Delta HNCO$ decreasing linearly ($R^2 = 0.36$) with increasing modified combustion efficiency for the MOYA-II fires. Consequently, Fig. 8b shows methane emission factor decreasing with lower $\Delta HCN/\Delta HNCO$ ratio. This further affirms that difference in combustion completeness is the primary driver of methane EF variability observed during MOYA-II. Unfortunately, A similar analysis could not be carried out for MOYA-I as the ToF-CIMS was not fitted to the aircraft during the MOYA-I flights.

As in flight C132, $N_2O$ measurements for flight C133 were unreliable and data were discarded due to the effects of aircraft motion on the instrument optical bench during turbulence. Furthermore, issues with the temperature control of the QCLAS optical bench meant that the baseline noise and drift of the $N_2O$ signal increased during this flight. This resulted in a reduced signal-to-noise ratio of $N_2O$ in the plume. For these reasons, an $N_2O$ EF is not reported for flight C133. However, optical bench temperature control was adequate during flight C134, and aircraft turbulence did not impact $N_2O$ data quality significantly during sampling of some fire plumes. Hence calculation of $N_2O$ EFs was possible for six of the nine fire plumes sampled during flight C134.

In general, the $N_2O$ mixing ratio enhancements in the fire plumes are small (<10 ppb) relative to the background variability (and instrumental noise) of the $N_2O$ dataset (up to 2 ppb). Hence the signal-to-noise ratios of the in-plume $N_2O$ enhancements are poorer than the in-plume enhancements of other species. As a result of this, the uncertainty relative to the mean $N_2O$ EF for C134 is larger than those seen in the other species measured. Despite the combination of instrument issues and poor signal-to-noise ratio, the $N_2O$ EF for flight C134 ($0.08 \pm 0.01$ g kg$^{-1}$) agrees within overlapping uncertainty with the savannah fire $N_2O$ EF reported by Andreae (2019) ($0.17 \pm 0.09$ g kg$^{-1}$)

Fig. 7 shows strong linear relationships between MCE and $CH_4$, $CO_2$ and CO EFs for both MOYA-I and MOYA-II. There is no discernible linear relationship between the $N_2O$ EFs from C134 and MCE, which is shown in the supplementary information in Fig. S5. It is worth noting that $CH_4$ EFs and corresponding MCE for the far-field flights C006 and C007 are not included in Fig. 7, as the EFs from these flights are representative of multiple fires with a mixture of phases, whereas the near-field EFs are representative of single fires with a single combustion efficiency associated with them. This trend is expected as higher MCE, and hence more complete flaming combustion, would lead to increased emission of more oxidised combustion products ($CO_2$) and less emission of more reduced compounds such as $CH_4$. Despite this, $CH_4$ EFs measured in Uganda appear to be significantly higher than those measured in Senegal at the same MCE, hence methane emissions from the Ugandan wildfires sampled appear to be higher, and this difference is independent of combustion efficiency. The difference in the linear regressions could possibly be accounted for by differences in the Senegalese and Ugandan fuel mixtures. However, due to detailed analysis of the fuel burned in this study being impossible, and with the likelihood of the fuel source being mixed, the

effect of differing fuel content is difficult to quantify. An additional hypothesis is that higher average soil moisture in northern Uganda compared to south-west Senegal could result in soil parching and consequent release of methane-rich air from the soil surrounding wildfires, however more work is required to investigate if soil moisture could affect wildfire methane EFs in this way.

## 3.2 Far-Field sampling

Flights C006 and C007 were designed to characterise the regional continental outflow of air masses influenced by biomass burning from Senegal and wider West Africa. C006 and C007 involved sampling at various altitudes from 16 m above sea level (ASL) to 6500 m ASL over the West African Atlantic coastline. C006 involved straight-and-level runs directly west of the Casamance region of Senegal targeted during the near-field flights C004 and C005. A strong measured easterly wind indicated continental outflow from the southwest Casamance region of Senegal during flight C006. Sampling during flight C007 was conducted further south, running parallel to the coastline of Guinea-Bissau due to the more complex meteorology encountered during the flight.

In order to identify the approximate origin and age of the biomass burning emissions sampled during the far-field flights, the National Oceanic and Atmospheric Administration (NOAA) HYbrid Single-Particle Lagrangian Integrated Trajectory (HYSPLIT) model was used to calculate 3-dimensional single-particle back-trajectories of air masses sampled during C006 and C007 (Stein et al. 2015). HYSPLIT back-trajectories were run at 60 second intervals during times where biomass burning emissions were sampled (Supplementary: Fig. S6 and S7) The back-trajectories for C006 shown in Fig. 9a and 9b indicate that the age of the biomass burning plumes sampled was approximately 8 hours. Furthermore, the sampled air mass appeared to have advected over the south-western Casamance region, with the highest CO concentrations observed in air masses that passed directly over this region. Thus, the sampled outflow represents a well-mixed air mass influenced by the fire regions targeted in the near-field. The HYSPLIT back-trajectories for C007 shown in Fig. 9c and 9d highlight the much more complex atmospheric dynamics influencing the sampled air masses during flight C007 as opposed to C006. The biomass burning emissions sampled during C007 originated from Guinea-Bissau, Guinea, Sierra Leone, and South Western Senegal, all of which were undergoing active burning during this time as shown in Fig 1a. With these complex air masses, the approximate age of the biomass burning emissions observed in C007 was estimated to be older than that in C006, with an approximate age of 1-2 days. Consequently, the emissions sampled in C007 were representative of a wider area of West African biomass burning than C006, spanning from south-west Senegal down to Sierra Leone. Due to the significantly older plume age of the C007 biomass burning emissions, it is possible that significant chemical aging and/or mixing of background air with plume air has occurred, and hence the ERs or EFs derived from this flight may not be representative of single source regions (see Sect. 2.6).

Box-whisker altitude profiles for flight C006 and C007 are shown in Fig. 10. Fig. 10c shows peak CO concentrations in air masses at approximately 1600 m ASL during flight C006. This is consistent also with fire plume injection heights observed during near-field sampling. Both $CH_4$ and $CO_2$ altitude profiles in Fig. 10a and 10b also show enhanced concentrations up to approximately 1600 m ASL, with a rapid decrease in mean CO concentration from 2000 m ASL, indicating free tropospheric air above this. This was confirmed by analysis of measured thermodynamic profiles (not shown in this work). The altitude profiles in Fig. 10d-f show that during flight C007, peak CO concentration as well as the highest mean CO concentration was measured at approximately 1400 m ASL. Concurrently, $CH_4$ and $CO_2$ mixing ratios were enhanced up to approximately 3400 m ASL. Above this, CO, $CH_4$ and $CO_2$ mixing ratios decreased to background free tropospheric concentrations with comparatively small ranges. In comparison to flight C006, in C007 the biomass burning emissions appeared to be more mixed throughout the polluted boundary layer.

A linear weighted regression was fitted to data points for $CH_4$ and $CO_2$ versus tracer-CO (Fig. 11) for samples within the biomass burning plume, using a statistical CO threshold to identify the smoke plumes from fires (as described in Sect. 2.6). The gradient of this fit was equivalent to the ERs with respect to CO and included in Table 1. Fig. 11a and 11b show strong linear trends between in-plume $CH_4$ and $CO_2$ vs CO for flight C006. with $R^2$ values of 0.70 and 0.76 respectively.

Although some degree of linearity is identifiable, the observed trends shown in Fig. 11c and 11d are significantly weaker than those seen for flight C006, with $R^2$ values of 0.14 for $CH_4$ vs CO and 0.49 for $CO_2$ vs CO. The higher variance in the C007 linear regressions, when compared with C006, could be attributed to mixed phases of burning and/or mixed degrees of chemical aging present within the same biomass burning influenced air mass. Therefore homogenisation of species from individual fire areas within the whole enhanced plume in C007 may be incomplete, and multiple fire phases with distinct combustion efficiencies or plume aging may explain the poorer fits seen in C007.

As observed in Sect. 3.1 in the near-field sampling flights C004 and C005, the methane EF calculated for C006 ($1.6 \pm 0.18$ g $kg^{-1}$) and C007 ($2.4 \pm 0.25$ g $kg^{-1}$) is more comparable to savannah and grassland burning methane EF ($2.7 \pm 2.2$ g $kg^{-1}$ reported by Andreae. (2019). This is attributed to the mixed nature of the fuel source, likely comprising of facultatively deciduous forest litter and savannah grasses

MCE values of $0.94 \pm 0.0041$ for C006 and $0.96 \pm 0.0037$ for C007 are also shown in Table 1. It is likely that biomass burning signatures with a higher smouldering component were sampled in C006, which is further evidenced by the lower $CO_2$ EFs determined for C006. In contrast, the $CH_4$ EF is higher for C007, in which more complete combustion is inferred from the MCE. It is expected that this is due to the aging of species sampled offshore in a recirculated airmass in C007 (as shown Fig. 12), and hence an indication that ERs and EFs may not be representative of the source fires. Despite ERs and EFs being shown

for C007 in Fig. 11c and 11d, the EFs for C007 are not included in the mean calculation for Senegalese biomass burning EFs.

## 4. Conclusion

Airborne observations of $CH_4$, $CO_2$, and CO emissions from biomass burning were carried out in southern Senegal in February/March 2017 and northern Uganda in January 2019. Mean EFs of $1.8 \pm 0.19$ g kg$^{-1}$ for $CH_4$, $1633 \pm 171.4$ g kg$^{-1}$ for $CO_2$ and $67 \pm 7.4$ g kg$^{-1}$ for CO were obtained from the Senegalese fires, with a mean modified combustion efficiency of 0.94

$\pm 0.005$. Mean EFs of $3.1 \pm 0.35$ g kg$^{-1}$ for $CH_4$, $1610 \pm 169.7$ g kg$^{-1}$ for $CO_2$ and $78 \pm 8.9$ g kg$^{-1}$ for CO were obtained for the Ugandan fires, with a mean modified combustion efficiency of $0.93 \pm 0.004$. A mean $N_2O$ EF of $0.08 \pm 0.01$ g kg$^{-1}$ is also reported for six fire plumes sampled over Uganda. $CH_4$ EFs showed strong linear relationships with modified combustion efficiency for both Senegal and Uganda. The variability in EFs within each study area was attributed to the mixed-phase nature of the fires, with a range of combustion efficiencies observed. These results also suggest that Ugandan fires have a higher

methane emission factor for the equivalent combustion efficiency observed for Senegal. This may be a consequence of the difference in fuel between the Ugandan savannah grass and cropland waste fuels, and the Senegalese forest litter and grassland fuel. This highlights the importance of considering both regional and local variability when attempting to spatially scale biomass burning emissions, and suggests that singular regional EF values may lead to inaccurate estimates. Further work to constrain EFs at more local scales and for more specific (and quantifiable) fuel types will serve to improve global estimates of

biomass burning emissions of climate-relevant gases.

This work demonstrates the value of airborne measurements for characterising biomass burning emissions from multiple fires over wide areas. This study has provided unique *in situ* datasets in two geographical regions where there has hitherto been little study by aircraft measurement. The results will improve understanding of the role of African biomass burning in the

680 global carbon budget, and the work demonstrates the importance of good knowledge of fuel mixture for the accurate reporting of EFs. This study demonstrates the utility of airborne measurements for characterising biomass burning emissions from multiple fires over wide areas. Further work is required to investigate the link that fire fuel content may have on the emission of methane from biomass burning.

## Data availability

FAAM ARA data from the MOYA project can be found on the CEDA archive (http://archive.ceda.ac.uk/) at https://catalogue.ceda.ac.uk/uuid/d309a5ab60b04b6c82eca6d006350ae6 (FAAM, NERC, Met Office. 2017).

**Author Contribution**

Patrick Barker: Lead author, Data curation, formal analysis, methodology, investigation, validation, writing – original draft, writing - review and editing. Grant Allen: Principal Investigator, project administration, conceptualisation, supervision, funding acquisition, validation, writing – original draft, writing – review and editing. Martin Gallagher: Validation, supervision, writing – review and editing. Joseph R. Pitt: Data curation, formal analysis, validation, supervision. Rebecca E. Fisher: Data curation, formal analysis, investigation, writing – review and editing. Thomas Bannan: Data curation, formal analysis, writing – review and editing. Euan G. Nisbet: Principal investigator, project administration, conceptualisation, funding acquisition, validation, writing – review and editing. Stephane J. -B. Bauguitte: Data curation, formal analysis, validation, writing – review and editing. Dominika Pasternak: Data curation, formal analysis, validation, writing – review and editing. Samuel Cliff: Data curation, formal analysis, validation. Marina B. Schimpf: Data curation, formal analysis, validation. Archit Mehra: Data curation, formal analysis, writing – review and editing. Keith N. Bower: project administration, conceptualisation. James D. Lee: project administration, conceptualisation, writing – review and editing. Hugh Coe: conceptualisation, validation. Carl J. Percival: Data curation, validation, writing – review and editing.

**Competing interests**

The authors declare no conflict of interest.

**Acknowledgements**

The data used in this publication has been collected as part of the Methane Observations and Yearly Assessments (MOYA) project funded by the Natural Environment Research Council (NERC) (The Global Methane Budget, University of Manchester reference: NE/N015835/1 Royal Holloway, University of London reference: NE/N016211/1). Airborne data was obtained using the BAe-146-301 Atmospheric Research Aircraft (ARA) flown by Airtask Ltd and managed by FAAM Airborne Laboratory, jointly operated by UK Research and Innovation (UKRI) and the University of Leeds. We would like to give special thanks to the Airtask pilots and engineers and all staff at FAAM Airborne Laboratory for their hard work in helping plan and execute successful MOYA project flights. We acknowledge the use of MODIS data and imagery from LANCE FIRMS operated by NASA's Earth Science Data and Information System (ESDIS) with funding provided by NASA Headquarters (doi: 10.5067/FIRMS/MODIS/MCD14DL.NRT.006). The maps used in Fig. 1 and Fig. 5 are obtained from ArcGIS software (Sources: Esri, HERE, Garmin, Intermap, Increment P Corp., GEBCO, USGS, FAO, NPS, NRCAN, GeoBase, IGN, Kadaster NL, Ordnance Survey, Esri Japan, METI, Esri China (Hong Kong). © OpenStreetMap contributors and the GIS User Community). The maps used in Fig. 9 are produced using the Python Matplotlib Basemap package, using Global Self-consistent, Hierarchical, High-resolution Geography Database (GSHHG) coastline and border data (Wessel and Smith. 1996). P. A. Barker is in receipt of a PhD studentship as part of the NERC Earth, Atmosphere and Ocean Doctoral Training Partnership (EAO DTP) (NERC grant reference: NE/L002469/1)

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

**Table 1: Mean CH₄, CO₂, N₂O and CO EFs and MCE for all MOYA-I (Senegal) and MOYA-II (Uganda) fires sampling flights. Both the standard error on the mean and the mean measurement uncertainty (MU) for EFs and MCEs during each flight for each species is also given. EFs and MCEs compiled from studies of tropical forest and savannah biomass burning by Andreae. (2019) are also shown. All EFs are reported in units of g kg⁻¹. \*Note that N₂O EFs could only be calculated for six of the nine fire plumes sampled during flight C134.**

|  | Flight No | N | CH₄ mean | SE | MU | CO₂ mean | SE | MU | CO mean | SE | MU | N₂O mean | SE | MU | MCE mean | SE | MU |
|---|---|---|---|---|---|---|---|---|---|---|---|---|---|---|---|---|---|
| MOYA-I | C004 | 7 | 2.3 | 0.13 | 0.24 | 1612 | 3.4 | 170 | 84 | 2.3 | 8.7 | - | - | - | 0.93 | 0.0047 | 0.0031 |
| | C005 | 12 | 1.4 | 0.11 | 0.15 | 1647 | 4.3 | 174 | 61 | 2.9 | 6.2 | - | - | - | 0.95 | 0.0024 | 0.0030 |
| | C006 | | 1.6 | - | 0.18 | 1625 | - | 170 | - | - | - | - | - | - | 0.94 | | 0.0041 |
| | C007 | | 2.4 | - | 0.25 | 1663 | - | 173 | - | - | - | - | - | - | 0.96 | - | 0.0037 |
| MOYA-II | C132 | 2 | 5.2 | 0.15 | 0.55 | 1554 | 4.0 | 164 | 109 | 2.3 | 11.3 | - | - | - | 0.90 | 0.0021 | 0.0042 |
| | C133 | 11 | 2.8 | 0.21 | 0.30 | 1620 | 7.0 | 171 | 72 | 2.6 | 7.4 | - | - | - | 0.94 | 0.0038 | 0.0041 |
| | C134 | 9 | 3.1 | 0.70 | 0.22 | 1609 | 23.9 | 174 | 79 | 14.0 | 8.1 | 0.08* | 0.01 | 0.01 | 0.93 | 0.0128 | 0.0042 |
| Andreae (2019) | Tropical Forest | | 6.5 | - | 1.6 | 1620 | - | 70 | 104 | - | 39 | - | - | - | 0.91 | - | 0.03 |
| | Savannah and Grassland | | 2.7 | - | 2.2 | 1660 | - | 90 | 69 | - | 20 | 0.17 | - | 0.09 | 0.94 | - | 0.02 |

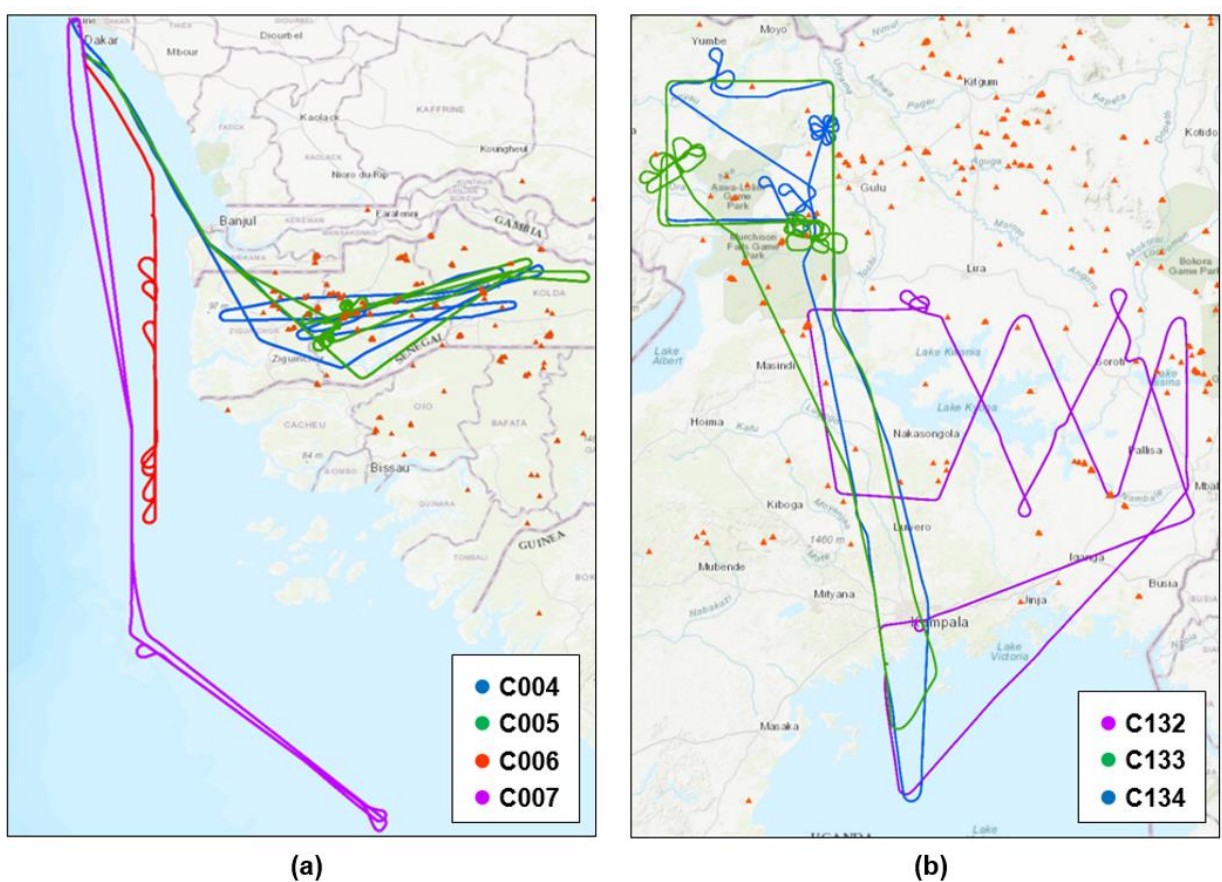

**(a)**  **(b)**

**Figure 1: FAAM ARA flight tracks of (a) MOYA-I biomass burning sampling flights C004 (Blue), C005 (Green), C006 (Red) and C007 (Purple) over the south western region of Senegal and the Atlantic seaboard. and (b) MOYA-II biomass burning sampling flights C132 (Purple), C133 (Green) and C134 (Blue) over northern Uganda. MODIS infrared satellite retrievals of fires present between (a) 28 February 2017 and 02 March 2017 and (b) 28 January 2019 and 29 January 2019 are also shown (orange triangles). © OpenStreetMap contributors and the GIS User Community 2020. Distributed under a Creative Commons BY-SA License.**






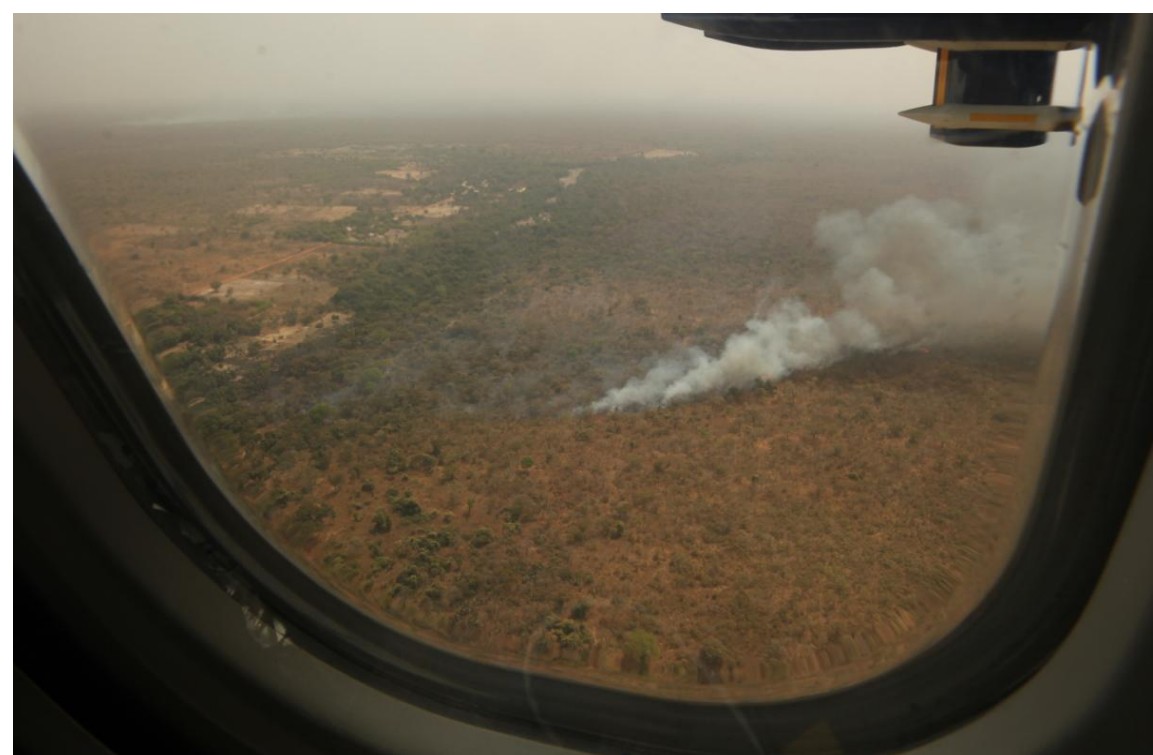

**Figure 2: Photograph of Senegalese wildfire taken from aboard the FAAM ARA during flight C005 of the MOYA-I campaign**

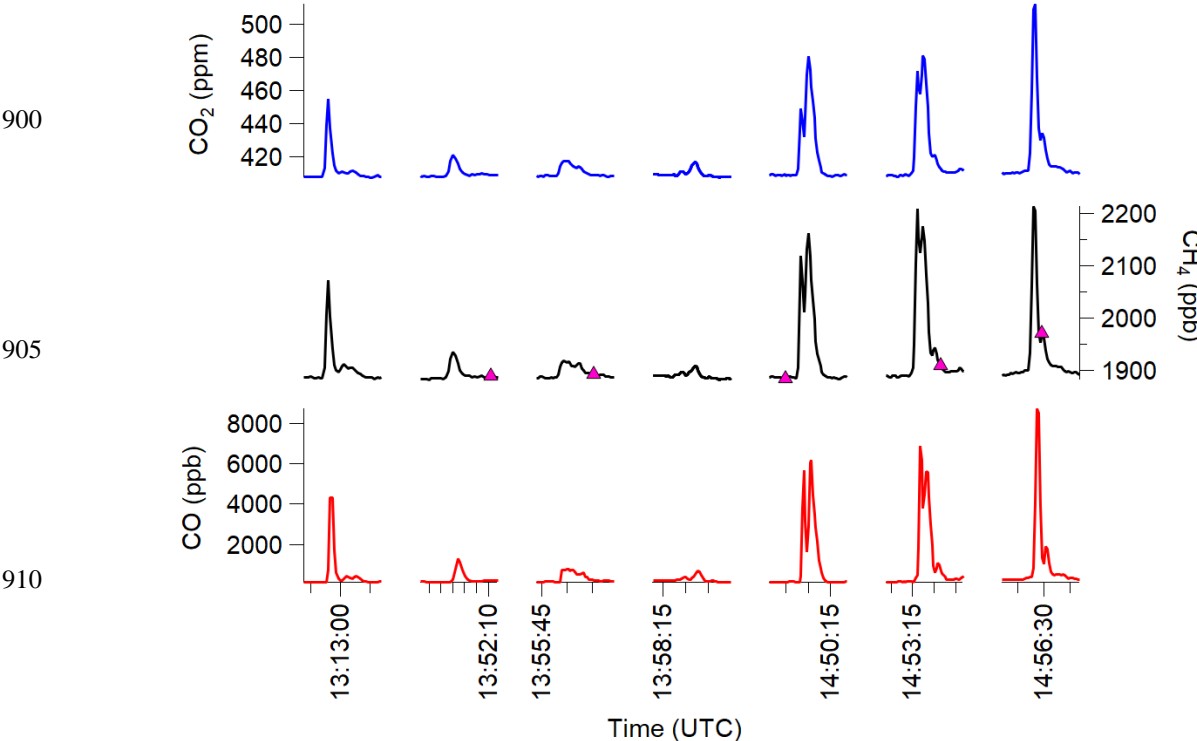

**Figure 3: time series of CO (red), CH₄ (black), CO₂ (blue) and concentrations in the plumes analysed for flight C005. Median WAS canister fill times are marked on the CH₄ time series as pink triangles. Note that some WAS taken in background regions are not shown here.**








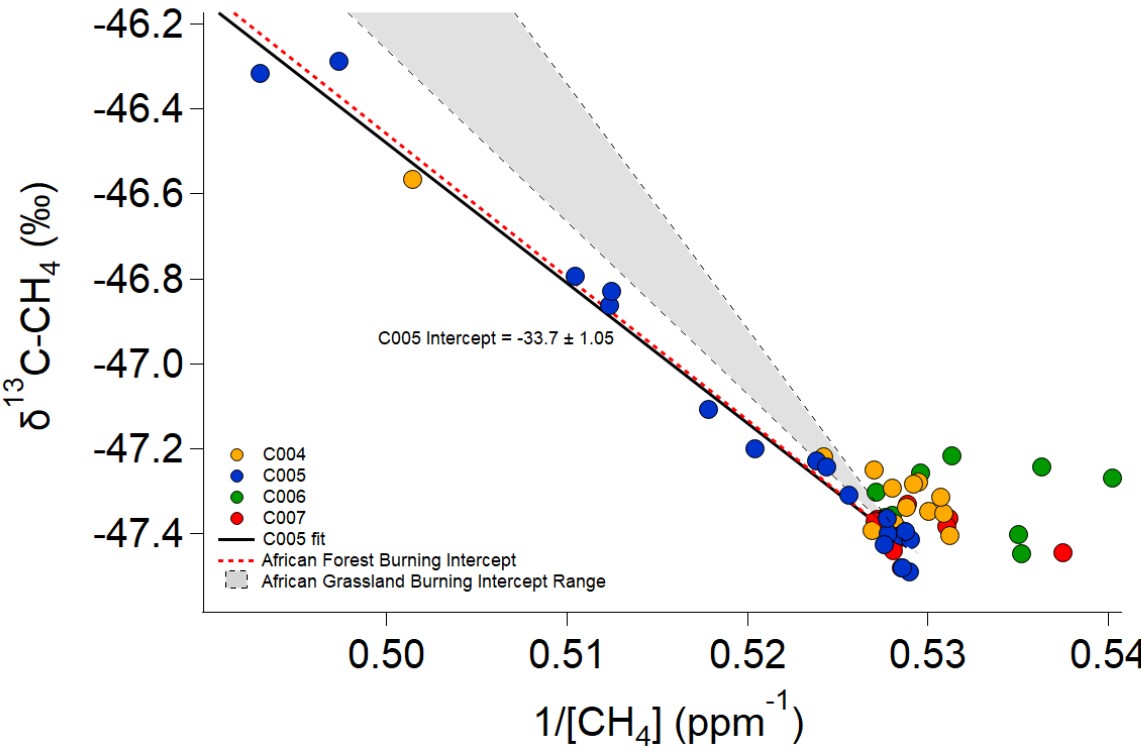


**Figure 4: Keeling plot (δ¹³C-CH₄ vs inverse CH₄ mixing ratio) for all flights in the MOYA-I (Senegal) analyses. A linear fit of points from flight C005 (blue) is also displayed. Simulated fits of African forest (red dashed line) and grassland (grey shaded area) burning using the intercepts and intercept ranges reported by Chanton et al. (2001) are also shown.**


(a)                                                (b)

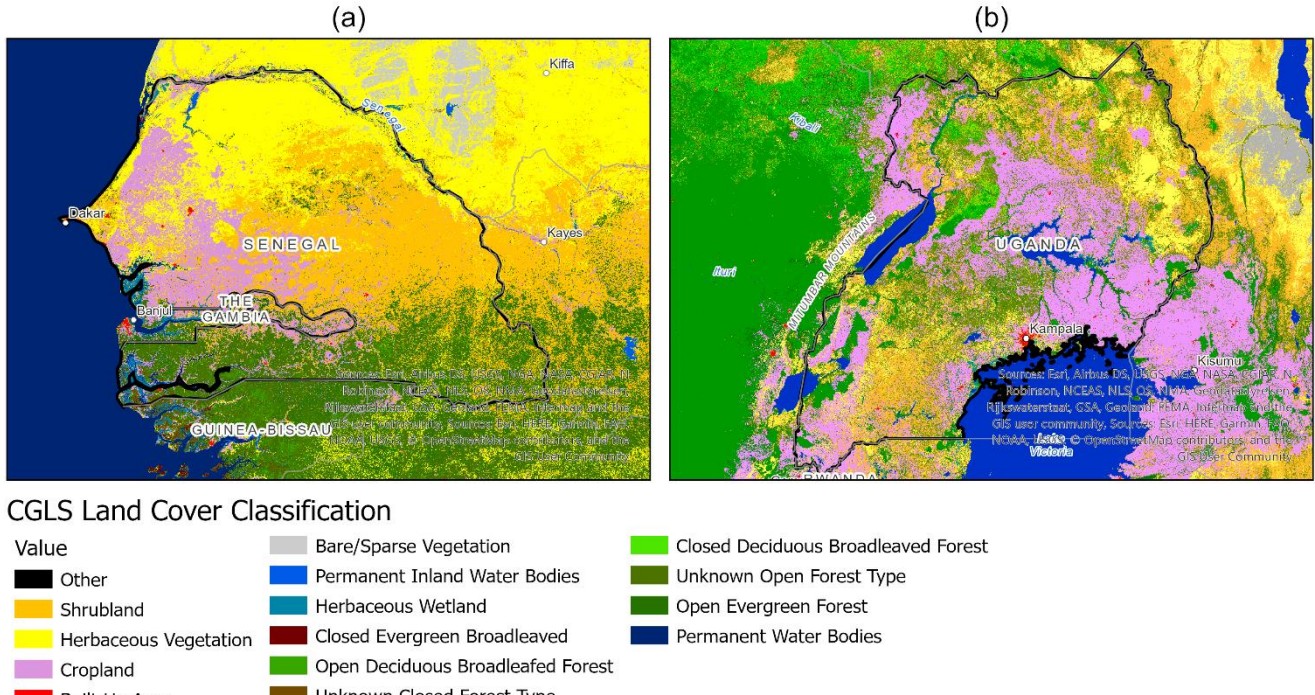

CGLS Land Cover Classification

Value
- ■ Other
- ■ Shrubland
- ■ Herbaceous Vegetation
- ■ Cropland
- ■ Built-Up Area

- ■ Bare/Sparse Vegetation
- ■ Permanent Inland Water Bodies
- ■ Herbaceous Wetland
- ■ Closed Evergreen Broadleaved
- ■ Open Deciduous Broadleafed Forest
- ■ Unknown Closed Forest Type

- ■ Closed Deciduous Broadleaved Forest
- ■ Unknown Open Forest Type
- ■ Open Evergreen Forest
- ■ Permanent Water Bodies

**Figure 5: (a) Land cover classification map of Uganda from 2019. (b) Land cover classification map of Senegal from 2017. Data is obtained from the Copernicus Global Land Service Africa Land Cover Maps, which are derived from PROBA-V satellite observations (Buchhorn et al. 2019). © OpenStreetMap contributors and the GIS User Community 2020. Distributed under a Creative Commons BY-SA License.**





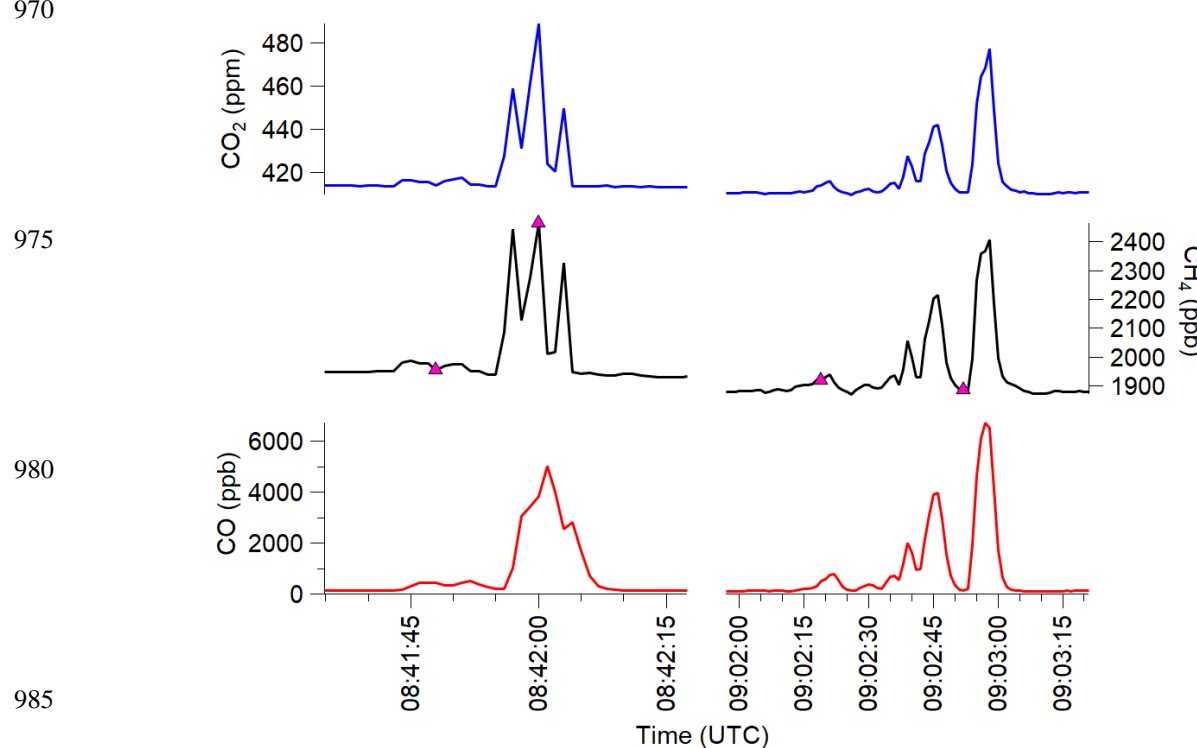


**Figure 6: time series of CO (red), CH₄ (black), CO₂ (blue) and concentrations in the plumes analysed for flight C005. Median WAS canister fill times are marked on the CH₄ time series as pink triangles. Note that some WAS taken in background regions are not shown here.**

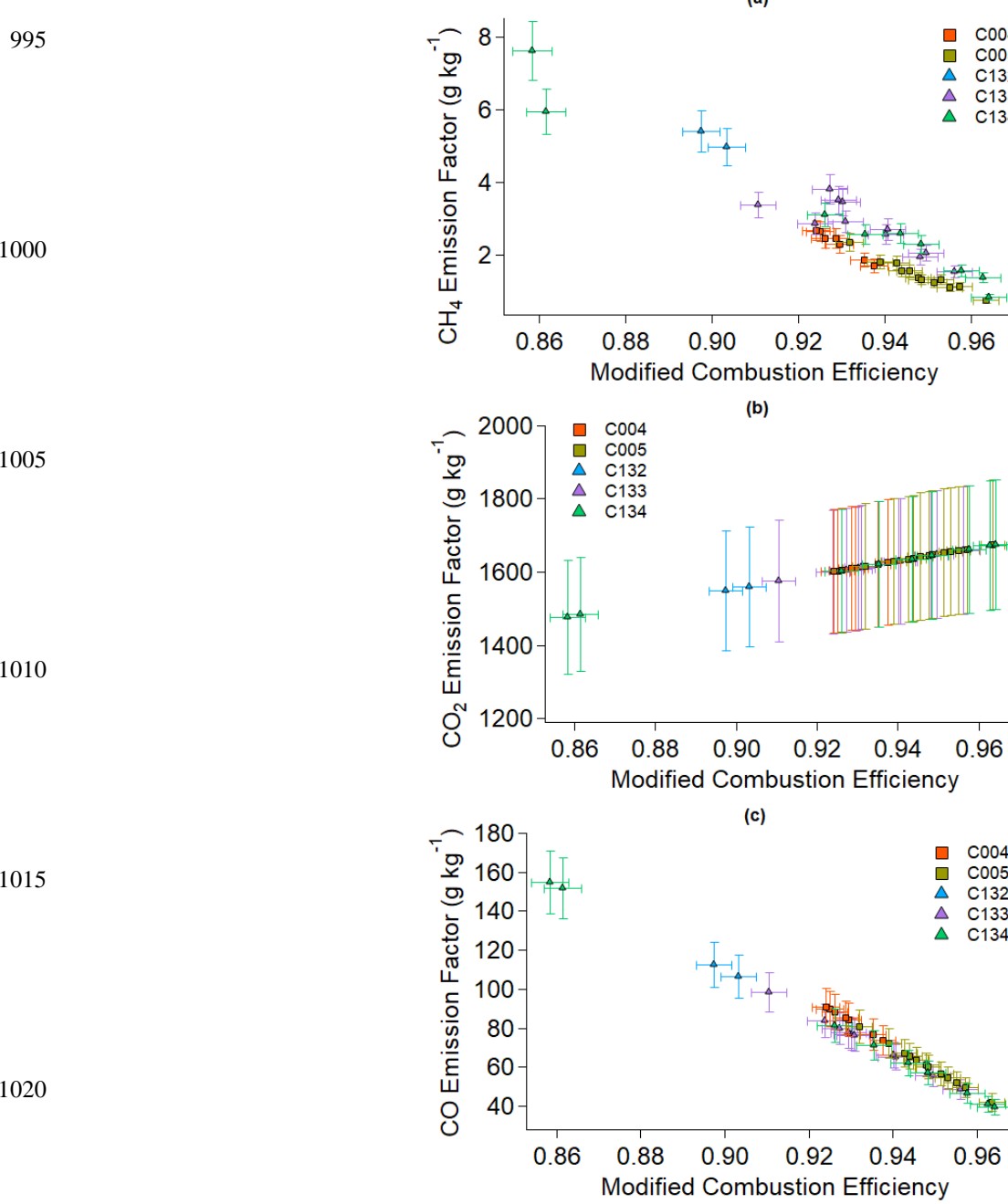

**Figure 7: (a) Methane, (b) CO₂ and (c) CO EF vs modified combustion efficiency for all biomass burning plumes sampled over all flights (squares are MOYA-I and triangles are MOYA-II). Points are coloured by flight number.**

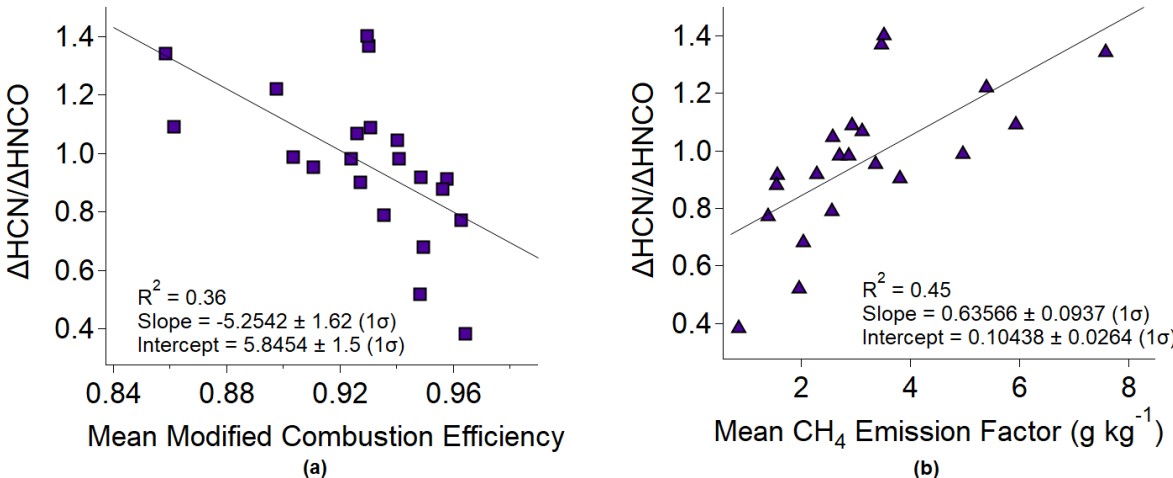

1030

**Figure 8: Plot of HCN enhancement over HNCO enhancement in biomass burning plumes vs (a) mean modified combustion efficiency and (b) mean methane EF in g kg$^{-1}$ for all MOYA-II data.**

1035

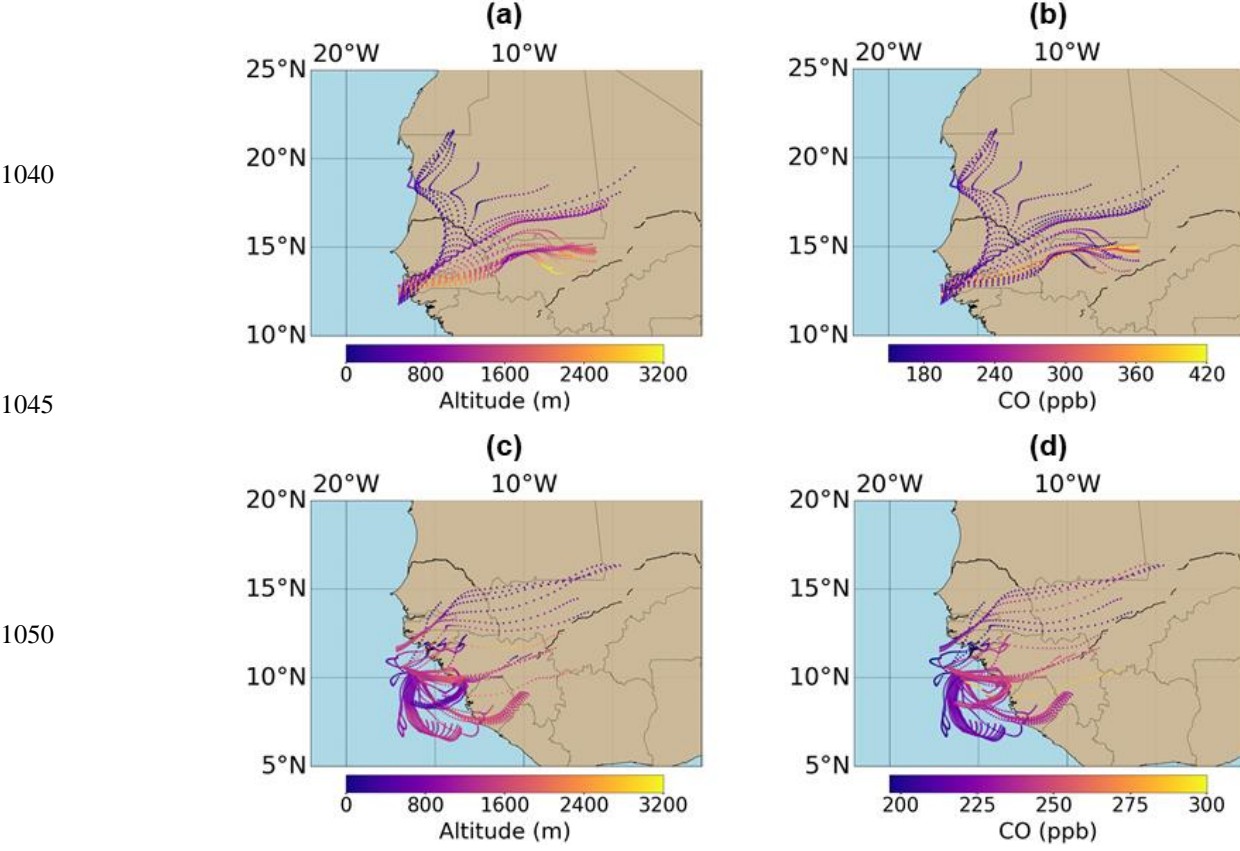

**Figure 9: 2-day HYSPLIT back-trajectories from sections of the flight tracks of flight C006 and C007 during which biomass burning emissions were sampled (the in fire plume data from Fig. 10). The back-trajectories are coloured by (a) trajectory altitude and (b) CO mixing ratio at the trajectory end-point on the flight C006 flight track. (c) and (d) show the back-trajectories for flight C007, coloured by trajectory altitude and CO mixing ratio respectively. Trajectories are run at 60 second intervals of in-plume flight data. The basemaps are obtained from Global Self-consistent, Hierarchical, High-resolution Geography Database (GSHHG) data (Wessel and Smith. 1996).**

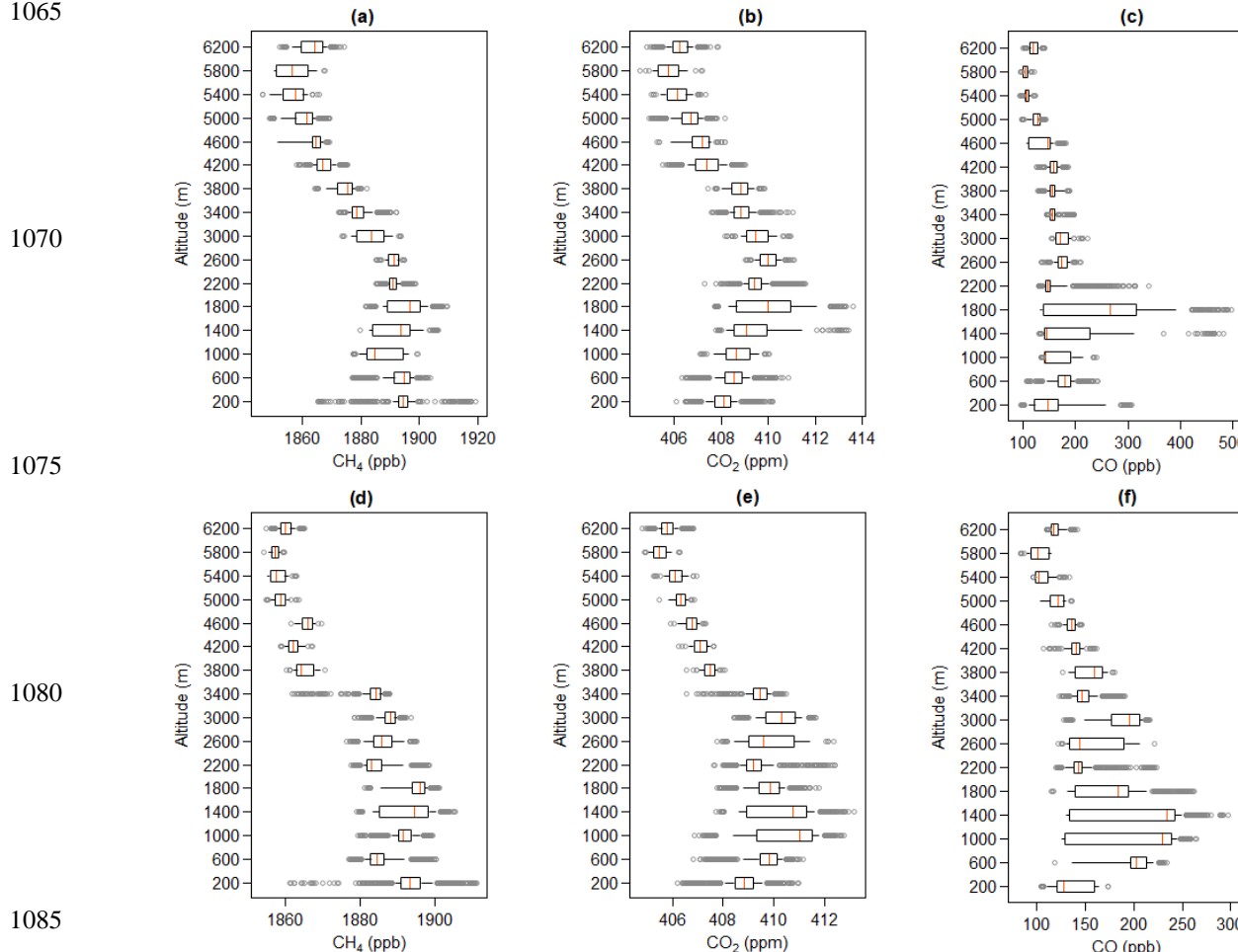

**Figure 10.** Box-whisker plots of (a) CH₄, (b) CO₂ and (c) CO altitude profiles for flight C006 and (d) CH₄, (e) CO₂ and (f) CO altitude profiles for flight C007. Altitude is divided into 400 m vertical bins for all box-whisker plots. The boxes represent the 25th and 75th percentiles, whiskers represent 10th and 90th percentiles, and the grey circular points are outliers.

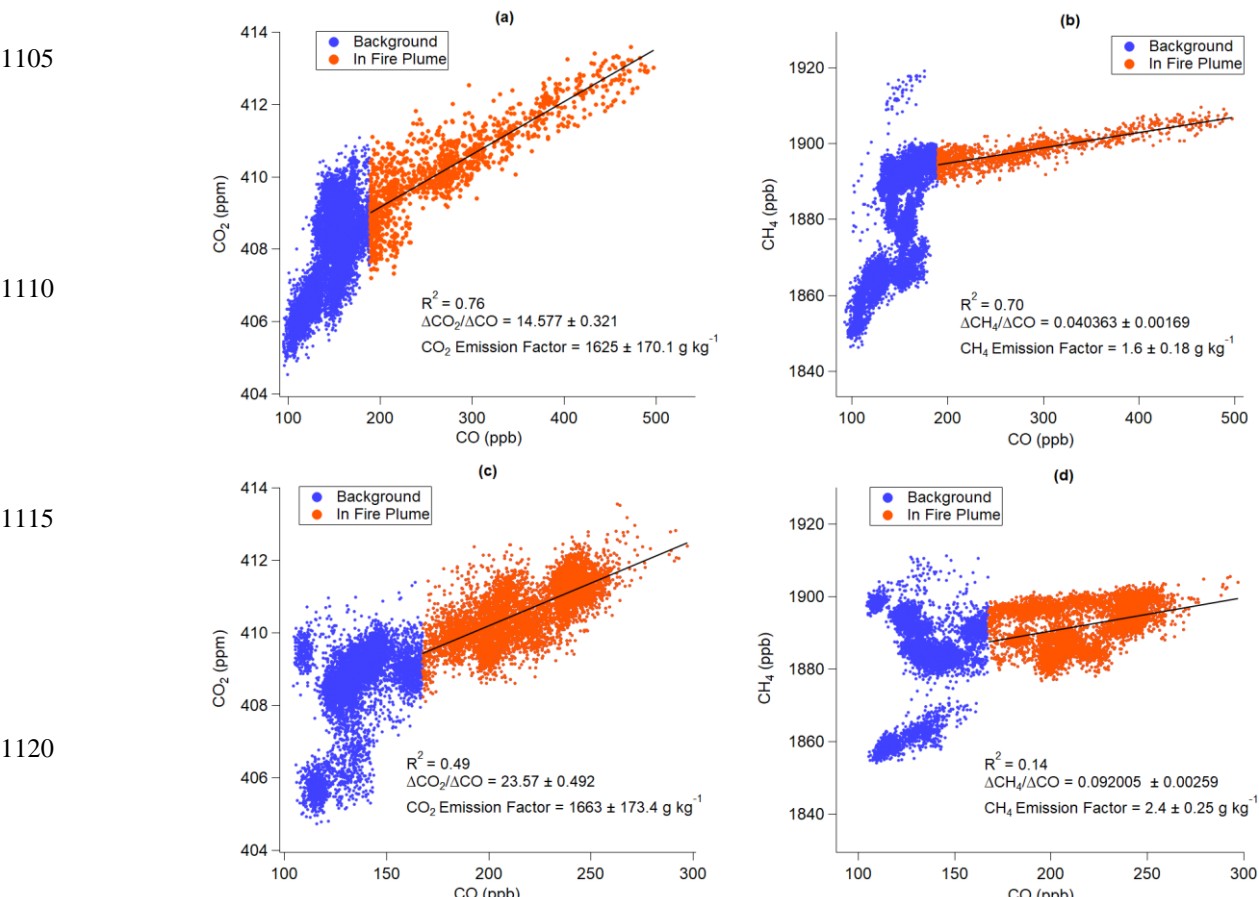

**Figure 11: Linear regressions of in-plume (a) CH₄ and (b) CO₂ mixing ratio versus in-plume CO mixing ratio for flight C006 and (c) CH₄ and (d) CO₂ mixing ratio versus in-plume CO for flight C007. The linear regressions are calculated using the York regression method, and are weighted towards CO and CH₄/CO₂ measurement uncertainty (York et al. 2004). ERs obtained from the slope are also shown, as well as the calculated EFs.**
