# Peer review of "Airborne measurements of fire Emission Factors for African biomass burning sampled during the MOYA Campaign"

_Atmospheric Chemistry and Physics, 2020_

## Referee Comment (RC1) · Anonymous Referee #1 · 1 Aug 2020

General Comments:

This is a clearly written paper, easy to follow and understand, and presents an accessible and rigorous analysis of an important dataset. The authors nicely motivate their work and the measurement campaigns and lay out a clear story which is backed up logically. The figures are generally good and easy to understand. The primary weakness in the paper is the use of the ordinary least squares (OLS) regression, there are some figure selection issues, and some minor technical questions. With minor revisions this paper will be a welcome addition to the literature.

Specific Comments:

[Figure]

This paper uses OLS regression, which is inappropriate for calculating trace gas ratios. While OLS is scale invariant, ratios taken from the slope of this regression cannot be inverted. Since all OLS regressions presented in this paper use observed data, a weighted regression (such as the York regression; please see York et al. 2004, Cantrell 2008, and Wu and Yu 2018, references below) using instrument precision as the variable weights will provide a robust trace gas ratio that is both scale invariant and slope invertible. Please do not use unweighted orthogonal distance regression (ODR) in place of OLS for trace gas ratios, as ODR is sensitive to the scales of the axes.

The emission factors calculated in this paper use a single value for the mass fraction of the dry fuel. With the context that the primary fuel is changing from C3 to C4 plants between MOYA-I and MOYA-II, is this single mass fraction appropriate for both regions?

I have some question about whether figure 6 is currently adding significantly to the discussion. In its current formulation this figure is only applicable to the MOYA-II flights, and the B panel is repeating information that it shown with better context in Figure 1. If a similar ground cover map for Senegal was available keeping figure 6 as a ground cover discussion would be useful, but in its current form figure 6 could be moved to the supplementary material without impacting the discussion.

For figures 7 and 8, the plots where clearly motivated in the text; I would like to see this analysis applied to the other trace gas species. While the primary motivation of these missions was to quantify methane emissions, extending this analysis to CO, CO2, and the additional MOYA-II species either in these plots or as supplemental material would be a welcome addition to the paper.

The Far Field Sampling section feels moderately unorganized compared to the rest of section 4. Currently the paper discusses the two flights separately, with 6 total figures; changing the structure to discussing both flights together by type of analysis (back trajectories, vertical distributions, and trace gas ratios) would allow the authors to combine the plots into three figures and compare and contrast the two flights more

effectively.

Technical Corrections and Small Issues:

There are some minor structural issues the methods sections. Section 2 and 3 would work better as a combined methods section, and I recommend splitting both the CIMS instrument and the WAS measurements into their own individual subsections.

Was the data used in this paper time synced? This is important for capturing trace gas ratios from regression methodology and should be included in the methods.

Since the identification of the plumes was done with statistics above the background, how was the "width" of the plume assigned? Is the plume only valid when your tracer species is at or above the threshold, or is the plume assigned with some buffer time on either side?

Figure 2 is out of order – it's cited after Figures 3 and 4.

In Figure 4 was the linear relationship applied to all measurements in the C004 and C005 flights, or did it leave off the high inverse CH4 data? If some data was left off, what was the threshold for that decision?

The following statement in lines 480-485: "The trends in mean MCE and EFs observed during C132, C133, and C134 suggest that EFs are mostly determined by the completeness of combustion over other factors." would be more effectively shown as a figure, either in the supplementary material or as a replacement for figure 6.

The time series plots should include notations or icons showing when the WAS samples were collected. Please include time series plots for the Far Field Sampling flights; including them in the supplementary material would be fine.

For the Far Field Sampling figures, two things need to be clarified in either the text or the figure captions. First, are the high split back trajectories run on distinct plumes, or where they initiated on a regular time step? And second, what is the vertical binning

on the box and whisker plots in Figures 10 and 13?

Weighted Regression References: York, D., Evensen, N. M., MartÄśÌĄnez, M. L., & De Basabe Delgado, J. (2004). Unified equations for the slope, intercept, and standard errors of the best straight line. American Journal of Physics, 72(3), 367–375. https://doi.org/10.1119/1.1632486

Cantrell, C. A. (2008). Technical Note: Review of methods for linear least-squares fitting of data and application to atmospheric chemistry problems. Atmospheric Chemistry and Physics, 8(17), 5477–5487. https://doi.org/10.5194/acp-8-5477-2008

Wu, C., & Yu, J. Z. (2018). Evaluation of linear regression techniques for atmospheric applications: the importance of appropriate weighting. Atmospheric Measurement Techniques, 11(2), 1233–1250. https://doi.org/10.5194/amt-11-1233-2018

Weighted Regression followup note: Since it appears that the plots in the paper were created in Igor Pro, a weighted regression that is equivalent to the York Regression (assuming uncorrelated errors between the variables) can be calculated using the built in curvefit function. In this formulation, the uncertainty waves are the pointwise precision (or uncertainties) for the x and y waves in the same units as x and y (ie, ppb, ppm). /I=0 (the default) sets the regression to expect reciprocal uncertainties, which can cause issues, so make sure to check before running. I am happy to iterate about weighted regressions in the comments if you have any questions.

Curvefit/odr=2 line y_data_wave /x=x_data_wave/I=1 /Xw=x_unc_wave /W=y_unc_wave
* * *

---

## Referee Comment (RC2) · Anonymous Referee #2 · 15 Aug 2020

The authors present results from two airborne studies performed in 2017, and 2019 over Senegal and Uganda, respectively, in order to investigate fire emission factors from African biomass burning plumes. The manuscript is well written and easy to follow. Comments above are mostly suggestions in order to improve the visualization of the graphs and present the comparisons to other studies in a clearer manner.

Comments

Line 196: Please discuss possible instrument interferences of N2O.

Line 250: Overflowing with what? How does someone correct for humidity changes? Is there an ambient air catalyst converter for this purpose? It would be informative to

include an inlet setup for the CIMS in the SI for the reader to be able to follow the details if they feel the need to.

Line 410: It should be mentioned somewhere in the manuscript that methane was synchronized to the canister start stop times. Was it by calculating averages or medians or integrated signal?

Line 411: The lesser extent is only supported by 1 data point. Only C005 shows a consistent trend. No suggestion can be made for the other flights. Please make it clear in the text. Also, in Fig. 4 it would improve the comparison if the authors add the intercepts from previous studies to the graph as horizontal lines with their uncertainty as shaded background. This way the comparison to previous studies will be better supported.

Line 416: The authors suggest that in Uganda the Keeling plot analysis could not be used. Could the authors show a Figure of this in the SI in comparison to Senegal?

Line 421: I wouldn't consider this a forest rather than a combined forest & grassland area based on this one picture (the quality of the picture is not great). This is also mentioned by the authors later in the text. Please rephrase and consistently mention throughout the manuscript.

Line 419-440: Even if the emissions are a mixture of forest and grassland I don't understand why this wouldn't be consistent for both the isotopic analysis and the EFs. The influence of grassland burning is reflected both in the MCEs and EFs but not the isotopic ratios. What is the isotopic intercept difference of C3 forest litter and C4 tropical grasses and maize? Is the difference dramatic? I think the discussion in this section could be improved by mentioning all the parameters that can affect both the isotopic ratio and EF calculations and then conclude which one is expected to play a dominant role in the observed differences. Also, could these differences be related to different parts of the plume being sampled from the different inlets (CH4 vs Canisters) at different parts of the plain? How far away were the two instrument inlets? If the plumes are

not dense could this become an issue?

Line 575: Visual observations don't strongly support this.

Calculation of the slope to CO based on a linear fit is how the authors determine the ERs for the far-flights. It will improve comparisons if the authors apply this approach for the other flights and determine whether they observe any substantial differences compared to equation 1.

Figure comments

Figure 3: I would recommend restructuring the Fig. in order to make it more instructive. Some suggestions below: Exclude (a) and keep only (b) this way you have a longer timeseries for the reader to look at. Split x-axis to 7 axes that are plotted against the same y-axis. Each x-axis will show an individual plume crossing (in total 7 plumes, 7 x-axes) zoomed in to the respective plume and their backgrounds. Please do the same for Fig. 5 and all the other flights in the SI.

Figure 6: It would be informative if the flight tracks are shown in one of the two maps. This graph could also be moved to the SI.

Figure 7: Would the ratio of HCN / HNCO change depending on the particle humidity and therefore the more efficient uptake to the particles of HNCO? I guess it will not be significant compared to the emission differences but maybe an interesting topic to discuss here.

Figure 10: Make the extreme markers smaller and grey.

Technical comments

Line 259: Maybe delete FIGAERO-CIMS instrument analysis software and just include the ARI Tofware version 3.1.0. Readers may be confused when reading FIGAERO and expect particle-phase measurements.

Line 297: delete "an".

[Figure]

Line 301-302: How do you know if the flights display no significant plume ageing? Please specify based on the airborne measurements.

Line 302: delete "confidently".

Line 424: Change "Mean" to "mean".

Line 428: Change "It" to "it".

Line 437: Delete ".".

Line 457: There is inconsistent use of brackets here and at other parts of the manuscript.

Line 459: Delete ".".

Line 460: Change "," to ".".
* * *

---

## Author Comment (AC1) · 9 Oct 2020

**Author response to referee comments**

We would like to thank both reviewers for their thorough and constructive comments on our manuscript and for recognising the positive contribution the paper can make to understanding of the nature of biomass burning emissions in Africa. We believe that implementation of the suggestions made by the referees has greatly improved the presentation of the revised paper. Referee comments are addressed in turn below. The original comments are coloured red and author comments coloured black.

**Referee 1:**

**Specific Comments:**

This paper uses OLS regression, which is inappropriate for calculating trace gas ratios. While OLS is scale invariant, ratios taken from the slope of this regression cannot be inverted. Since all OLS regressions presented in this paper use observed data, a weighted regression (such as the York regression; please see York et al. 2004, Cantrell 2008, and Wu and Yu 2018, references below) using instrument precision as the variable weights will provide a robust trace gas ratio that is both scale invariant and slope invertible. Please do not use unweighted orthogonal distance regression (ODR) in place of OLS for trace gas ratios, as ODR is sensitive to the scales of the axes.

We agree that the use of OLS regression may not be optimal for the determination of trace gas emission ratios as was done in the original manuscript. York et al., 2004, regressions have now been implemented in place of the OLS fits in Figure 11 following the reviewer's suggestion. Very slight changes in final emission factors and their respective uncertainties are now seen compared with the original OLS approach.

The emission factors calculated in this paper use a single value for the mass fraction of the dry fuel. With the context that the primary fuel is changing from C3 to C4 plants between MOYA-I and MOYA-II, is this single mass fraction appropriate for both regions?

This potential difference was indeed considered in previous iterations of the manuscript, but estimating fuel carbon fractions for each study area in lieu of a more detailed knowledge of the fuels was not possible. We were unable to find literature carbon fractions that were more representative of the study areas/fuel mixtures encountered during this work. Previous aircraft biomass burning studies such as Yokelson et al. 2009 have also used a single carbon fraction value for fires originating from different fuels, but have included a ±10% uncertainty range to account for potential error due to this fuel uncertainty. To be consistent with previous studies, we have now implemented a ±10% uncertainty into our carbon fraction to account for differences in fuel carbon content between the study areas. This uncertainty is propagated through to the final emission factors presented in the revised paper. The difference between methane emissions (between different areas) remains significant (outside of error) after this change.

I have some question about whether figure 6 is currently adding significantly to the discussion. In its current formulation this figure is only applicable to the MOYA-II flights, and the B panel is repeating information that it shown with better context in Figure 1. If a similar ground cover map for Senegal was available keeping figure 6 as a ground cover discussion would be useful, but in its current form figure 6 could be moved to the supplementary material without impacting the discussion.

We acknowledge that Figure 6 is a relatively weak figure in its original format and does not add much to the discussion. The MODIS fire map of Uganda has now been replaced with a land cover classification map of Senegal, and references and discussion have been added for both land cover classifications.

For figures 7 and 8, the plots where clearly motivated in the text; I would like to see this analysis applied to the other trace gas species. While the primary motivation of these missions was to quantify methane emissions, extending this analysis to CO, CO2, and the additional MOYA-II species either in these plots or as supplemental material would be a welcome addition to the paper.

This is an excellent idea. We have added plots of $CO_2$ and CO emission factor versus modified combustion efficiency to the updated manuscript. A $N_2O$ emission factor vs modified combustion efficiency plot is also included in the SI, where no linear relationship was observed.

The Far Field Sampling section feels moderately unorganized compared to the rest of section 4. Currently the paper discusses the two flights separately, with 6 total figures; changing the structure to discussing both flights together by type of analysis (back trajectories, vertical distributions, and trace gas ratios) would allow the authors to combine the plots into three figures and compare and contrast the two flights more effectively.

We agree. This section did not flow well in its original form, and feels repetitive with separation of the same figure types into individual flights. The back-trajectories, vertical box-whisker plots and trace gas ratios have now been combined from six figures into three separate figures, with each figure encompassing both flights. The discussion text has also been reorganised to discuss by analysis type, rather than by flight.

**Technical Corrections and Small Issues:**

There are some minor structural issues the methods sections. Section 2 and 3 would work better as a combined methods section, and I recommend splitting both the CIMS instrument and the WAS measurements into their own individual subsections.

We agree. This has now been changed. The two sections have been combined and instrumental methods are now divided into the subsections '$CH_4$, $CO_2$, CO and $N_2O$ Instrumentation', 'HCN and HNCO Instrumentation (Chemical Ionisation Mass Spectrometer)' and 'Whole Air Sampling and Methane Isotopic Analysis'

Was the data used in this paper time synced? This is important for capturing trace gas ratios from regression methodology and should be included in the methods.

All aircraft instrumentation is synced to a time server on-board the aircraft, and WAS capture start and end times are recorded using the time from this time server. This information has now been added to the revised methods section.

Since the identification of the plumes was done with statistics above the background, how was the "width" of the plume assigned? Is the plume only valid when your tracer species is at or above the threshold, or is the plume assigned with some buffer time on either side?

Despite using a statistical concentration threshold to select plumes, identifying a 'catch all' single method to assign plume width and background areas for all plumes proved difficult due to the individual systematic differences between plumes and area above the threshold. Near-field plume start and end time, as well as background regions, were assigned manually using the data. The statistical threshold was only used to automate and identify the plumes for further analytical treatment. This has now been clarified in the 'Calculation of Emission Ratios and Emission Factors' subsection.

Figure 2 is out of order – it's cited after Figures 3 and 4.

Thank you. This has been corrected.

In Figure 4 was the linear relationship applied to all measurements in the C004 and C005 flights, or did it leave off the high inverse CH4 data? If some data was left off, what was the threshold for that decision?

Upon further examination of the WAS times, some WAS points have now been discarded from the analysis due to samples being taken on the ground or too close to Dakar (potentially influenced by urban emissions) on the profile ascent out of the airport to the study area. The linear fit is now applied to all remaining points of the C005 WAS samples. We have decided to remove the singular C004 point from the fit, as a single point is not useful in constraining a linear relationship. The updated Figure 4 is now included in the manuscript

The following statement in lines 480-485: "The trends in mean MCE and EFs observed during C132, C133, and C134 suggest that EFs are mostly determined by the completeness of combustion over other factors." would be more effectively shown as a figure, either in the supplementary material or as a replacement for figure 6.

We agree that this would be much clearer if illustrated in a figure. The strong linear relationship between EF and MCE is shown clearly in figure 7, which is coloured by each flight. We think a separate EF vs MCE graph for the mean EFs and MCEs for C132-C134 would essentially be a simplified version of figure 7, so fig. 7 has now been referenced in the statement highlighted by the reviewer.

The time series plots should include notations or icons showing when the WAS samples were collected.

We agree. These have been added.

Please include time series plots for the Far Field Sampling flights; including them in the supplementary material would be fine.

This is a good idea. These have been added to the supplementary information.

For the Far Field Sampling figures, two things need to be clarified in either the text or the figure captions. First, are the high split back trajectories run on distinct plumes, or where they initiated on a regular time step? And second, what is the vertical binning on the box and whisker plots in Figures 10 and 13?

The back-trajectories are simulated for 60 s intervals during times when biomass burning emissions were observed and sampled (from grey shaded areas in Fig. S5 and S6 in supplementary). The box-whisker plots use 400 m altitude bins. This has now been clarified in the text and captions.

**Referee 2:**

Line 196: Please discuss possible instrument interferences of $N_2O$.

While there are no obvious spectroscopic interferences of other species with the $N_2O$ spectral absorption feature, the nature of the $N_2O$ instrument issues were perhaps inadequately described in the original manuscript. The Aerodyne QCLAS instrument exhibits a significant known cabin pressure dependency, in which significant changes in altitude, and hence cabin pressure, change the refractive index in the open path section of the laser beam. This can lead to static optical fringes moving across the spectral baseline of the instrument, introducing both long-term drift and short-term artefacts into the $N_2O$ mole fraction data. Also, sharp changes in aircraft roll angle in tight turns can also introduce short-term artefacts as forces acting on optical components cause slight changes in alignment. A further issue encountered solely during the MOYA-II campaign was occasional loss of optical bench temperature control due to the high temperatures experienced within the aircraft during some flights.

Despite these known issues, the $N_2O$ plumes from which EF could be calculated were sampled at constant altitude with wings-level at constant optical bench temperature. So the instrument issues detailed above are expected to have no systematic effect on data quality. More information on these issues has now been added to the text.

Line 250: Overflowing with what? How does someone correct for humidity changes?
Is there an ambient air catalyst converter for this purpose? It would be informative to include an inlet setup for the CIMS in the SI for the reader to be able to follow the details
if they feel the need to.

We apologise for the lack of clarity in the text here. Background determinations were completed by overflowing $N_2$ gas just above the moveable critical orifice prior to the IMR, and humidity corrections were applied during post processing of the data, as described in the manuscript. Humidity dependency was determined by varying the IMR humidity while delivering a constant mixing ratio HCN in into the instrument over a range of atmospherically relevant concentrations, which is common practice.

More details have now been added to the text to explain the CIMS design. However, we believe it is not useful to include a diagram due to its simplicity and similarity to the setup of Lee et al. (2018), which is referenced and has a figure included in that study. This is now referenced in the text.

Line 410: It should be mentioned somewhere in the manuscript that methane was synchronized to the canister start stop times. Was it by calculating averages or medians
or integrated signal?

We apologise for the lack of clarity in the text here. The WAS canisters were subsequently analysed for mean $\delta^{13}C$ methane isotope ratio (GC-MS) and mean methane mole fraction (Picarro 1303 Cavity Ringdown IR spectrometer) in the lab. The inverse $CH_4$ mole fraction used in the Keeling plots is from the lab analysis of the WAS canisters, not from the in situ methane measurements aboard the aircraft. Each point in the Keeling plots is the mean $\delta^{13}C$-$CH_4$ and mean $1/[CH_4]$ from one canister. Lab and in situ mean $CH_4$ mole fraction during each WAS sampling period were compared and found to agree within uncertainty (not shown). The use of lab $CH_4$ instead of in situ $CH_4$ for Keeling plots has been made clearer in the text.

Line 411: The lesser extent is only supported by 1 data point. Only C005 shows a consistent trend. No suggestion can be made for the other flights. Please make it clear in the text. Also, in Fig. 4 it would improve the comparison if the authors add the intercepts from previous studies to the graph as horizontal lines with their uncertainty as shaded background. This way the comparison to previous studies will be better supported.

Thank you for this suggestion. We agree. This has now been clarified in the revised text. The intercepts from Chanton et al. (2000) are now included; an intercept range of -17 ‰ to -26 ‰ for African grassland burning is shown as a shaded area and an intercept of -30 ‰ for African forest burning is shown as a red dashed line. These simulated fits originate from the start point of the C005 fit.

Line 416: The authors suggest that in Uganda the Keeling plot analysis could not be used. Could the authors show a Figure of this in the SI in comparison to Senegal?

This is a useful idea – thank you. This has been added to the SI.

Line 421: I wouldn't consider this a forest rather than a combined forest & grassland area based on this one picture (the quality of the picture is not great). This is also mentioned by the authors later in the text. Please rephrase and consistently mention throughout the manuscript.

Agreed. This has been corrected and the suspected mixed nature of the fuel has now been made consistent throughout the manuscript

Line 419-440: Even if the emissions are a mixture of forest and grassland I don't understand why this wouldn't be consistent for both the isotopic analysis and the EFs. The influence of grassland burning is reflected both in the MCEs and EFs but not the isotopic ratios. What is the isotopic intercept difference of C3 forest litter and C4 tropical grasses and maize? Is the difference dramatic? I think the discussion in this section could be improved by mentioning all the parameters that can affect both the isotopic ratio and EF calculations and then conclude which one is expected to play a dominant role in the observed differences. Also, could these differences be related to different parts of the plume being sampled from the different inlets (CH4 vs Canisters) at different parts of the plain? How far away were the two instrument inlets? If the plumes are not dense could this become an issue?

This section has been revised to highlight the parameters that affect isotopic ratio and methane EF more clearly. We do not believe differences in plume sampling between in situ $CH_4$ and WAS $CH_4$ are the cause for this, however. The plumes encountered during C004 and C005 involved sharp enhancements that in most cases lasted for <20 s. The WAS fill times during these flights were approximately 20 s and captured the majority of the plume in all in-plume WAS samples.

Line 575: Visual observations don't strongly support this.

We agree. This sentence has been removed.

Calculation of the slope to CO based on a linear fit is how the authors determine the ERs for the far-flights. It will improve comparisons if the authors apply this approach for the other flights and determine whether they observe any substantial differences compared to equation 1.

We agree that this would provide important intercomparison and validation for each ER determination method. As an extension of the referee's suggestion, both the linear regression method and the peak integration method have been applied to all flights to give a comprehensive comparison of the two methods. The two methods show very good agreement, with final $CH_4$ and $CO_2$ EFs agreeing comfortably within uncertainty for all flights. The far-field flight EF uncertainties calculated using the integration method are significantly larger than other uncertainty ranges due to the high variability of the background mole fraction and relatively low enhancement in the plume compared to those seen in the near-field flights. This analysis is now shown in the revised SI.

For final EF reporting, the integration method is still used for near-field flights as this gives ER and EF data for each individual plume, allowing the EF vs MCE analysis to be carried out and reported. Linear regression is still used for far-field ER calculation as the EF uncertainty is minimised.

Figure 3: I would recommend restructuring the Fig. in order to make it more instructive. Some suggestions below: Exclude (a) and keep only (b) this way you have a longer timeseries for the reader to look at. Split x-axis to 7 axes that are plotted against the same y-axis. Each x-axis will show an individual plume crossing (in total 7 plumes, 7 x-axes) zoomed in to the respective plume and their backgrounds. Please do the same for Fig. 5 and all the other flights in the SI.

We agree. Figures 3, 5 and those in the SI have now been updated to include split X-axes for each plume, with the plume area zoomed in.

Figure 6: It would be informative if the flight tracks are shown in one of the two maps. This graph could also be moved to the SI.

Figure 6 has now been revised to include a land cover classification map for Senegal as well as Uganda (see response to Reviewer #1 also). However, we feel that adding flight tracks to the land cover maps may obscure the land cover classifications, so these remain separate in the revised mauscript.

Figure 7: Would the ratio of HCN / HNCO change depending on the particle humidity and therefore the more efficient uptake to the particles of HNCO? I guess it will not be significant compared to the emission differences but maybe an interesting topic to discuss here.

This is an interesting idea for further investigation and discussion, but this is beyond the scope of the work presented here.

Figure 10: Make the extreme markers smaller and grey.

This has now been changed.

Line 259: Maybe delete FIGAERO-CIMS instrument analysis software and just include the ARI Tofware version 3.1.0. Readers may be confused when reading FIGAERO and expect particle-phase measurements

We agree with this. This has been changed.

Line 297: delete "an".

Changed. Thank you.

Line 301-302: How do you know if the flights display no significant plume ageing? Please specify based on the airborne measurements.

Plume ageing is inferred based on airmass history from HYSPLIT back trajectories from in-plume data for far-field flights as discussed in the text. All near-field flights sampled biomass burning emissions essentially at the source (directly above and confirmed visually), so no plume ageing was assumed. This has now been clarified in the text.

Line 302: delete "confidently".

Removed.

Line 424: Change "Mean" to "mean".

Changed.

Line 428: Change "It" to "it".

Changed.

Line 437: Delete ".".

Removed.

Line 457: There is inconsistent use of brackets here and at other parts of the manuscript.

Brackets have been made consistent throughout the manuscript.

Line 459: Delete ".".

Removed.

Line 460: Change "," to ".".

Changed.